# Behavioral health and experience of violence among cisgender heterosexual and lesbian, gay, bisexual, transgender, queer and questioning, and asexual (LGBTQA+) adolescents in Thailand

**Wit Wichaidit**[1,2]*, **Natnita Mattawanon**[3], **Witchaya Somboonmark**[4], **Nattaphorn Prodtongsom**[4], **Virasakdi Chongsuvivatwong**[1], **Sawitri Assanangkornchai**[1,2]

1 Faculty of Medicine, Epidemiology Unit, Prince of Songkla University, Hat Yai, Thailand, 2 Centre for Alcohol Studies, Hat Yai, Thailand, 3 Faculty of Medicine, Department of Obstetrics and Gynecology, Chiang Mai University, Chiang Mai, Thailand, 4 Faculty of Science, Division of Computational Science, Prince of Songkla University, Hat Yai, Thailand

* wit.w@psu.ac.th

## Abstract

### Background

Assessment of health disparities between population groups is essential to provide basic information for resource prioritization in public health. The objective of this study is to assess the extent that behavioral health outcomes and experience of violence varied between cisgender heterosexual adolescents and those who identified as lesbian, gay, bisexual, transgender, queer and questioning, and asexual (LGBTQA+) in the 5th National School Survey on Alcohol Consumption, Substance Use and Other Health-Risk Behaviors.

### Methods

We surveyed secondary school students in years 7, 9 and 11 in 113 schools in Thailand. We used self-administered questionnaires to ask participants about their gender identity and sexual orientation and classified participants as cisgender heterosexual, lesbian, gay, bisexual, transgender, queer and questioning, or asexual, stratified by sex assigned at birth. We also measured depressive symptoms, suicidality, sexual behaviors, alcohol and tobacco use, drug use, and past-year experience of violence. We analyzed the survey data using descriptive statistics with adjustment for sampling weights.

### Results

Our analyses included data from 23,659 participants who returned adequately-completed questionnaires. Among participants included in our analyses, 23 percent identified as LGBTQA+ with the most common identity being bisexual/polysexual girls. Participants who identified as LGBTQA+ were more likely to be in older year levels and attending general education schools rather than vocational schools. LGBTQA+ participants generally had

**Data Availability Statement:** All relevant data are within the paper and its Supporting Information files.

**Funding:** Funding for the Survey was provided by the Thai Health Promotion Foundation and the Center for Alcohol Studies (Award Number: 61-02029-0074, Recipient: Wit Wichaidit). Wit Wichaidit and Sawitri Assanangkornchai are salaried researchers at the Center for Alcohol Studies. The Thai Health Promotion Foundation had no role in study design, data collection and analysis, decision to publish, or preparation of the manuscript.

**Competing interests:** The authors have declared that no competing interests exist.

higher prevalence of depressive symptoms, suicidality, and alcohol use than cisgender heterosexual participants, whereas the prevalence of sexual behaviors, lifetime history of illicit drug use, and past-year history of violence varied widely between groups.

## Conclusion

We found disparities in behavioral health between cisgender heterosexual participants and LGBTQA+ participants. However, issues regarding potential misclassification of participants, limitation of past-year history of behaviors to the context of the COVID-19 pandemic, and the lack of data from youths outside the formal education system should be considered as caveats in the interpretation of the study findings.

## Introduction

Assessment of health disparities and experience of violence between groups and sub-groups in a given population is essential to prioritize the allocation of resources in behavioral health and health promotion. Such assessment is essential to identify priority groups for allocation of resources in behavioral health and health promotion.

One domain of disparities is those with regard to gender identifies and sexual orientations, e.g., disparities between cisgender-heterosexual persons compared to "lesbian, gay, bisexual, transgender, queer and question, and asexual" (LGBTQA+) persons. Gender identities and sexual orientations are diverse and exist on a spectrum in a given population [1–3]. The term "sex" refers to "the combinations of physical characteristics typical of males or females", whereas the term "gender identity" refers to "a person's internal sense of being male, female, or. . .a blend of both or neither" [4]. The term "assigned male at birth (AMAB)" refer to individuals "believed to be male when born and initially raised as boys", whereas the term "assigned female at birth (AFAB)" refers to individuals "believed to be female when born and initially raised as girls" [4]. The term "cisgender" refers to someone whose gender identity matches their sex assigned at birth, whereas the term "transgender" is an umbrella term [5], used as an adjective to refer to "someone whose gender identity doesn't match their sex assigned at birth" [4]. The term "transgender" may include those who identify in the male-female binary (e.g., transgender boys, transgender men, transgender girls, transgender women) as well as people who are non-binary persons. The term "sexual orientation" refers to the pattern of romantic or sexual attraction exhibited by a person to another person, and the term "asexual" refer to someone who lacks sexual attraction toward others, and "asexuality" can be regarded as either a sexual orientation or the lack thereof [6]. Thus, the term "cisgender-heterosexual" refers to the state in which a cisgender person is attracted to another cisgender person of the opposite sex assigned at birth, and the term "lesbian, gay, bisexual, transgender, queer and questioning, and asexual" (LGBTQA+) refers to any person who does not identify as cisgender-heterosexual. The plus (+) designation is used to represent those who do not identify as cisgender-heterosexual, but the components of the LGBTQA acronym themselves do not accurately reflect their identity [7].

Adolescence is the period in which people begin to develop a full understanding of their own gender identity and sexual orientation, but is also the age group where behavioral health issues impose a heavier burden relative to other health problems. LGBTQA+ people tend to have poorer behavioral health [8,9] and face greater level of physical and sexual violence than cisgender-heterosexual people [10,11]. LGBTQA+ adolescents have higher prevalence of

depression [9], suicidality [9], and substance abuse [10], violence and abuse [12], and social and human rights issues [11]. However, previous studies on behavioral health disparities in cisgender-heterosexual vs. LGBTQA+ adolescents were conducted in OECD countries, and findings may not be generalizable to low and middle-income countries which have vastly different socioeconomic and cultural contexts. In Thailand, a middle-income country in Southeast Asia, draft legislations on marriage equality and life partnerships have recently passed the first round in parliament [13]. However, incidents of violence against LGBTQA+ youths still occur [14] alongside systemic discrimination [11]. Furthermore, studies on LGBTQA+ health in Thailand tend to be small-scaled and focus only on specific sub-groups, such as gay men and transgender women [11]. Other groups, such as gender non-conforming women who are subjected to unique and pervasive forms of stigma [15], are scarcely studied, including with regard to their behavioral health. A previous nationally representative study only assessed disparities by gender identities but not sexual orientations [16], thus sexual minority groups were not represented in the findings.

Data from nationally-representative surveys of the general population of adolescents enable assessment of health disparities between cisgender-heterosexual vs. LGBTQA+ adolescents can provide relevant basic information for national-level stakeholders in adolescent health and LGBTQA+ rights. The objective of this study is to assess the extent that behavioral health outcomes and exposure to violence varied between cisgender-heterosexual and LGBTQA+ youths in Thailand's 5th National School Survey on Alcohol Consumption, Substance Use and Other Health-Risk Behaviors.

## Materials and methods

### Study design and participants

The 5th National School Survey on Alcohol Consumption, Substance Use and Other Health-Risk Behaviors (hence "Survey") was a cross-sectional survey conducted to provide information about the magnitude and trend of health risk behaviors among students in Thailand's formal education system. The participants included students in Year 7 (*Matthayom 1*), Year 9 (*Matthayom 3*), and Year 11 in the general education system (*Matthayom 5*) and the vocational education system (Vocational Certificate Year 2). The fifth Survey was conducted from November 2020 to March 2021 and included 24,143 students studying at 113 schools in 21 (out of 77) provinces of Thailand and 1 district of Bangkok, the capital of Thailand.

Thailand follows the 6-3-3 education system and all students begin Year 1 (Prathom 1) during the year that they are to reach 7 years of age [17]. The system requires students to repeat a grade level only in the most extreme circumstances [18]. Thus, the majority of participants in Year 7 were 12–13 years of age, participants in Year 8 were 14–15 years of age, and participants in Year 11 were 16–17 years of age. We included all students present in the sampled classroom on the day of data collection. We excluded students in Year 8, 10, and 12 in order for our study data to cover as broad an age range as possible given the existing statistical power.

### Ethical considerations

The Survey was approved by the Human Research Ethics Committee of the Faculty of Medicine, Prince of Songkla University (approval number: REC.63-446-18-2). During the information process and prior to giving verbal informed consent, students were reminded that all answers were optional and that they were free to stop filling the questionnaire at any time without any consequence. A waiver of the need for written informed consent for minors was approved by the Human Research Ethics Committee, based on the ground that requirement of parental information and consent could potentially preclude students in precarious domestic situation from reporting their information.

## Procedure

The investigators selected provinces in each of the 12 education regions (plus one district in the capital of Bangkok), then sampled the schools. Investigators then contacted the school administrators to ask for permission to conduct the Survey. During the weekly activity period or free period of a sampled classroom, trained enumerators visited the classroom. The enumerators requested the teacher or other school staff with disciplinary authority to leave the classroom briefly. The enumerators then introduced themselves, stated the objectives and procedures of the Survey, provided the students with an information sheet, distributed self-administered questionnaires, and asked the students for verbal consent to participate. Due to the sensitive nature of the Survey subject matters, trained enumerators did not record the name or other personally identifiable information of the participants during the study. Enumerators verbally confirmed the participants' understanding of information related to the study, including the participants' ability to refuse to answer any question and to withdraw from the study at any time. The participants then provided verbal consent and started completing the questionnaire. Enumerators also requested cooperation from the teacher and other school staff to refrain from looking at the participants' responses in order to maintain the participants' confidentiality and privacy. Prior to conducting the Survey, the investigators had received a waiver of documentation of consent from an institutional review board.

The participants then completed the questionnaires and immediately placed the questionnaires in individual envelopes. Members of the data entry team then performed data entry using EpiData Entry software. Participants who answered less than 70 percent of the questions in which skip patterns did not apply were considered to have submitted incomplete responses and were excluded from the analyses.

## Instrumentation: Assessment of gender identity and sexual orientation

We used two questions to assess gender identity and one question to assess sexual orientation. The first question to assess gender identity was "Sex at Birth" which contained two possible answer choices: "1) Male (*Dek Chai / Nai*)", and "2) Female (*Dek Ying / Nang Sao / Nang*)". The words in italic are titles written in Thai national identity cards and refers to the sex assigned at birth. The second question was "Gender Identity (with what gender do you actually identify?)" with five possible answer choices: "1) Male", "2) Female", "3) Gender diverse", "4) Not sure", and "9) Refuse to answer".

We assessed sexual orientation with one question: "To which gender are you attracted? (multiple answers allowed)". The possible responses were "1) Male"; "2) Female"; "3) Transgender female / *kathoey*"; "4) Transgender male / *tom*"; "5) Neither male nor female"; "6) I'm not sure to whom I am attracted"; "7) I'm not attracted to any gender", and; "9) Refuse to answer". The investigators did not include option 8 on the questionnaire. The investigators designated the value of 9 on many questions to indicate refusal to answer to facilitate the data entry process.

Based on findings of a previous study [16] and consultation among the investigators, we decided to classify the participants according to the criteria in *Table 1*. We excluded participants who did not provide answers to all three questions from our analyses.

Among the participants included in our subsequent analyses, there were differences between identity groups with regards to school type enrolled, year level of study, geographic region, weekly allowance, and grade point average (GPA) (*Table 2A and 2B*). Generally, participants at government schools and those in Matthayom 3 and Matthayom 5 more commonly identified as LGBTQA+ than participants at private schools and participants in Matthayom 1 or vocational schools. Supplementary analyses (*S1 Table in S5 File*) showed that study

**Table 1. Gender classification definitions of participants and proportions, weighted percent ± standard error (SE) (n = 21,323 participants).**

| Group | Definition | Percent ± SE* |
|---|---|---|
| 1) Cisgender heterosexual boys (n = 7,810 participants) | Cis-gender boys who were attracted to only cis-gender girls (Participants whose sex was "Male", gender identity was "Male", and reported to be attracted only to "Cis-Gender Female" persons) | 32.3%±2.0% |
| 2) Cisgender heterosexual girls (n = 8,225 participants) | Cis-gender girls who were attracted to only cis-gender boys (Participants whose sex was "Female", gender identity was "Female", and reported to be attracted only to "Cis-Gender Male" persons) | 34.9%±1.5% |
| 3) Cisgender boys, attracted only to cisgender boys ("Cisgender Homosexual Boy" or "Gay") (n = 496 participants) | Cis-gender boys who were attracted to only cis-gender boys (Participants whose sex was "Male", gender identity was "Male", and reported to be attracted only to "Cis-Gender Male" persons.) | 1.8%±0.3% |
| 4) Cisgender girls, attracted only to cisgender girls ("Cisgender Homosexual Girl" or "Lesbian") (n = 619 participants) | Cis-gender girls who were attracted to only cis-gender girls (Participants whose sex was "Female", gender identity was "Female", and reported to be attracted only to "Cis-Gender Female" persons.) | 2.2%±0.2% |
| 5) Cisgender Bisexual/Polysexual Boys (n = 199 participants) | Cis-gender boys who were attracted to more than one genders (Participants whose sex was "Male", gender identity was "Male", and reported to be attracted to more than one genders.) | 0.8%±0.1% |
| 6) Cisgender Bisexual/Polysexual Girls (n = 1,457 participants) | Cis-gender girls who were attracted to more than one genders (Participants whose sex was "Female", gender identity was "Female", and reported to be attracted to more than one genders.) | 6.9%±0.5% |
| 7) Transgender and gender diverse (TGD), assigned male at birth (AMAB) (n = 342 participants) | Persons assigned male at birth who identified as either "Female" or "Gender diverse" | 1.4%±0.1% |
| 8) Transgender and gender diverse (TGD), assigned female at birth (AFAB) (n = 395 participants) | Persons assigned female at birth who identified as either "Male" or "Gender diverse" | 2.0%±0.3% |
| 9) Asexual, assigned male at birth (AMAB) (n = 60 participants) | Persons assigned male at birth who reported their sexual orientation as solely "not attracted to any gender" | 0.3%±0.0% |
| 10) Asexual, assigned female at birth (AFAB) (n = 50 participants) | Persons assigned female at birth who reported their sexual orientation as solely "not attracted to any gender" | 0.2%±0.0% |
| 11) Otherwise queer and questioning, assigned male at birth (AMAB) (n = 433 participants) | Persons assigned male at birth who did not fit into any of the above categories | 1.9%±0.2% |
| 12) Otherwise queer and questioning, assigned female at birth (AFAB) (n = 1,237 participants) | Persons assigned female at birth who did not fit into any of the above categories | 5.6%±0.4% |
| 13) Incomplete information (n = 2,336 participants)* | Participants who did not answer question regarding sex or gender identity or sexual orientation and thus could not be placed in the categories above | 9.6%±1.1% |

*We excluded those who provided incomplete information from subsequent analyses; The labels in the quotation marks in the Group column are based only on the subjective interpretation of the authors.

**Table 2. a. Characteristics of study participants, weighted percent ± SE unless otherwise noted, part 1 (n = 21,323 participants) (ROW PERCENTS). b. Characteristics of study participants, weighted percent ± SE unless otherwise noted, part 2 (n = 21,323 participants).**

| Characteristic | Cisgender Heterosexual Boys (n = 7,810) | Cisgender Heterosexual Girls (n = 8,225) | Cisgender Homosexual Boys (n = 496) | Cisgender Homosexual Girls (n = 619) | Cisgender Bisexual / Polysexual Boys (n = 199) | Cisgender Bisexual / Polysexual Girls (n = 1,457) |
|---|---|---|---|---|---|---|
| **School type** | | | | | | |
| Government (n = 16,198) | 32.5% ± 0.0% | 39.5% ± 0.0% | 2.3% ± 0.0% | 2.3% ± 0.0% | 1.0% ± 0.0% | 8.8% ± 0.0% |
| Private (n = 5,125) | 44.5% ± 0.0% | 36.3% ± 0.0% | 1.5% ± 0.0% | 2.6% ± 0.0% | 0.8% ± 0.0% | 4.4% ± 0.0% |
| **Year Level** | | | | | | |
| Mathayom 1 (Year 7) (n = 6,160) | 40.2% ± 1.9% | 34.5% ± 1.4% | 1.4% ± 0.2% | 2.3% ± 0.3% | 0.8% ± 0.1% | 8.4% ± 0.6% |
| Mathayom 3 (Year 9) (n = 6,085) | 37.3% ± 2.6% | 37.1% ± 2.1% | 1.5% ± 0.3% | 2.3% ± 0.4% | 1.0% ± 0.2% | 7.3% ± 0.8% |
| Mathayom 5 (Year 11, General Education) (n = 5,607) | 25.6% ± 2.4% | 47.8% ± 2.5% | 3.2% ± 1.2% | 2.6% ± 0.4% | 1.1% ± 0.3% | 7.4% ± 0.4% |
| Vocational Certificate 2 (Year 11, Vocational Education) (n = 3,471) | 49.5% ± 1.1% | 30.9% ± 2.0% | 5.4% ± 1.4% | 3.7% ± 0.5% | 0.8% ± 0.3% | 2.9% ± 0.7% |
| **Religion** | | | | | | |
| Buddhism (n = 18,663) | 35.6% ± 2.2% | 37.9% ± 1.6% | 2.1% ± 0.4% | 2.5% ± 0.3% | 0.9% ± 0.2% | 8.0% ± 0.7% |
| Islam (n = 1,865) | 37.2% ± 6.0% | 46.7% ± 3.9% | 1.6% ± 0.2% | 2.1% ± 0.6% | 0.8% ± 0.2% | 3.9% ± 1.2% |
| Christianity (n = 544) | 37.9% ± 2.4% | 39.5% ± 3.7% | 0.8% ± 0.4% | 1.7% ± 0.5% | 0.8% ± 0.4% | 7.5% ± 1.5% |
| Others (n = 168) | 32.3% ± 3.7% | 14.3% ± 3.0% | 0.2% ± 0.2% | 1.6% ± 1.0% | 0.8% ± 0.6% | 10.4% ± 3.4% |
| **Region** | | | | | | |
| Special-Bangkok (n = 1,004) | 36.1% ± 3.6% | 31.1% ± 3.5% | 1.2% ± 0.3% | 1.8% ± 0.3% | 2.3% ± 0.4% | 10.8% ± 0.6% |
| Bangkok Metro Areas (n = 1,127) | 29.6% ± 5.8% | 44.5% ± 6.1% | 1.0% ± 0.0% | 1.3% ± 0.0% | 0.8% ± 0.2% | 9.7% ± 0.8% |
| Central (n = 5,029) | 35.6% ± 2.7% | 39.6% ± 2.3% | 1.3% ± 0.1% | 1.9% ± 0.2% | 0.5% ± 0.2% | 7.2% ± 0.4% |
| South (n = 4,468) | 40.0% ± 6.9% | 37.8% ± 5.0% | 1.9% ± 0.2% | 2.3% ± 0.3% | 1.2% ± 0.2% | 5.8% ± 0.7% |
| North (n = 5,759) | 34.4% ± 2.1% | 37.9% ± 2.1% | 1.2% ± 0.3% | 2.0% ± 0.2% | 0.7% ± 0.2% | 9.2% ± 0.6% |
| Northeast (n = 3,936) | 36.0% ± 2.7% | 39.9% ± 1.6% | 6.0% ± 1.9% | 5.0% ± 1.2% | 0.8% ± 0.2% | 4.8% ± 0.6% |
| **Living Situation**** | | | | | | |
| Family house/flat (n = 18,461) | 35.3% ± 2.0% | 39.2% ± 1.7% | 2.1% ± 0.4% | 2.5% ± 0.3% | 0.9% ± 0.1% | 7.6% ± 0.5% |
| School dorm (n = 622) | 47.5% ± 8.1% | 35.1% ± 4.8% | 0.7% ± 0.5% | 2.3% ± 0.9% | 0.4% ± 0.3% | 3.7% ± 1.4% |
| Outside dorm or others (n = 1,942) | 34.7% ± 3.0% | 36.5% ± 2.3% | 1.8% ± 0.3% | 2.0% ± 0.4% | 1.0% ± 0.4% | 9.3% ± 1.0% |
| Weekly allowance (THB) (mean ± standard errors) | 491.44±17.6 | 476.0±19.9 | 465.5±27.0 | 436.8±16.2 | 538.1±40.7 | 459.7±19.1 |
| **Grade point average (GPA)** | | | | | | |
| GPA = 0.1–1.0 | 81.5% ± 9.9% | 12.8% ± 9.0% | 2.2% ± 2.2% | 0.0% ± 0.0% | 0.2% ± 0.3% | 0.0% ± 0.0% |
| GPA = 1.1–2.0 | 63.2% ± 2.7% | 17.5% ± 1.6% | 4.4% ± 0.9% | 2.1% ± 0.6% | 1.0% ± 0.4% | 2.9% ± 0.6% |
| GPA = 2.1–3.0 | 46.3% ± 2.6% | 31.8% ± 2.1% | 2.7% ± 0.4% | 3.1% ± 0.4% | 1.2% ± 0.2% | 5.3% ± 0.6% |
| GPA = 3.1–4.0 | 27.4% ± 1.9% | 44.5% ± 1.6% | 1.6% ± 0.5% | 2.2% ± 0.3% | 0.8% ± 0.1% | 9.2% ± 0.6% |
| Unknown | 44.8% ± 4.0% | 31.1% ± 3.2% | 1.6% ± 0.6% | 1.6% ± 0.3% | 1.5% ± 0.4% | 6.7% ± 0.9% |

| Characteristic | TGD, AMAB (n = 342) | TGD, AFAB (n = 395) | Asexual, AMAB (n = 60) | Asexual, AFAB (n = 50) | Otherwise queer and questioning, AMAB (n = 433) | Otherwise queer and questioning, AFAB (n = 1,237) | P-value* |
|---|---|---|---|---|---|---|---|

*(Continued)*

| | | | | | | | p |
|---|---|---|---|---|---|---|---|
| **School type** | | | | | | | <0.001 |
| Government (n = 16,198) | 1.7% ± 0.0% | 2.5% ± 0.0% | 0.3% ± 0.0% | 0.3% ± 0.0% | 2.1% ± 0.0% | 6.8% ± 0.0% | |
| Private (n = 5,125) | 1.4% ± 0.0% | 1.5% ± 0.0% | 0.3% ± 0.0% | 0.2% ± 0.0% | 2.1% ± 0.0% | 4.3% ± 0.0% | |
| **Year Level** | | | | | | | <0.001 |
| Mathayom 1 (Year 7) (n = 6,160) | 1.2% ± 0.2% | 2.3% ± 0.4% | 0.4% ± 0.1% | 0.2% ± 0.1% | 1.9% ± 0.2% | 6.4% ± 0.8% | |
| Mathayom 3 (Year 9) (n = 6,085) | 1.7% ± 0.2% | 2.5% ± 0.4% | 0.2% ± 0.1% | 0.2% ± 0.1% | 2.4% ± 0.3% | 6.6% ± 0.5% | |
| Mathayom 5 (Year 11, General Education) (n = 5,607) | 2.1% ± 0.3% | 1.9% ± 0.4% | 0.3% ± 0.1% | 0.3% ± 0.1% | 2.0% ± 0.4% | 5.6% ± 0.6% | |
| Vocational Certificate 2 (Year 11, Vocational Education) (n = 3,471) | 1.3% ± 0.2% | 1.2% ± 0.4% | 0.2% ± 0.1% | 0.2% ± 0.1% | 1.7% ± 0.2% | 2.2% ± 0.5% | |
| **Religion** | | | | | | | <0.001 |
| Buddhism (n = 18,663) | 1.7% ± 0.1% | 2.2% ± 0.3% | 0.3% ± 0.0% | 0.2% ± 0.1% | 2.0% ± 0.2% | 6.4% ± 0.5% | |
| Islam (n = 1,865) | 0.5% ± 0.3% | 1.3% ± 0.8% | 0.3% ± 0.2% | 0.1% ± 0.1% | 2.0% ± 0.8% | 3.6% ± 0.4% | |
| Christianity (n = 544) | 1.0% ± 0.5% | 1.7% ± 0.5% | 0.9% ± 0.6% | 0.1% ± 0.1% | 1.3% ± 0.3% | 6.7% ± 1.6% | |
| Others (n = 168) | 2.9% ± 1.3% | 18.9% ± 4.7% | 0.8% ± 0.6% | 2.1% ± 1.1% | 10.1% ± 2.9% | 5.7% ± 2.5% | |
| **Region** | | | | | | | **0.005** |
| Special-Bangkok (n = 1,004) | 2.4% ± 0.1% | 3.0% ± 0.4% | 0.3% ± 0.0% | 0.2% ± 0.3% | 3.2% ± 0.1% | 7.5% ± 0.6% | |
| Bangkok Metro Areas (n = 1,127) | 1.0% ± 0.2% | 2.9% ± 0.7% | 0.0% ± 0.0% | 0.2% ± 0.0% | 2.2% ± 0.4% | 6.8% ± 0.2% | |
| Central (n = 5,029) | 1.7% ± 0.2% | 2.2% ± 0.3% | 0.3% ± 0.0% | 0.5% ± 0.1% | 1.9% ± 0.2% | 7.3% ± 0.2% | |
| South (n = 4,468) | 1.6% ± 0.3% | 2.4% ± 0.9% | 0.4% ± 0.1% | 0.1% ± 0.1% | 2.0% ± 0.4% | 4.6% ± 1.2% | |
| North (n = 5,759) | 1.6% ± 0.2% | 2.4% ± 0.3% | 0.4% ± 0.1% | 0.2% ± 0.0% | 2.2% ± 0.3% | 7.8% ± 0.6% | |
| Northeast (n = 3,936) | 1.5% ± 0.3% | 0.9% ± 0.2% | 0.2% ± 0.1% | 0.1% ± 0.0% | 1.7% ± 0.3% | 3.1% ± 0.5% | |
| **Living Situation**** | | | | | | | 0.115 |
| Family house/flat (n = 18,461) | 1.5% ± 0.1% | 2.3% ± 0.3% | 0.3% ± 0.0% | 0.2% ± 0.1% | 2.0% ± 0.2% | 6.1% ± 0.5% | |
| School dorm (n = 622) | 0.9% ± 0.6% | 0.7% ± 0.3% | 0.5% ± 0.3% | 0.3% ± 0.3% | 2.9% ± 1.7% | 5.0% ± 1.6% | |
| Outside dorm (n = 1,942) | 2.7% ± 0.5% | 2.1% ± 0.3% | 0.3% ± 0.1% | 0.1% ± 0.1% | 2.8% ± 0.6% | 6.7% ± 1.1% | |
| **Weekly allowance (THB) (mean ± standard errors)** | 566.9±31.4 | 476.1±17.0 | 482.2±46.3 | 468.6±72.0 | 498.1±27.7 | 461.6±20.9 | <0.001 |
| **Grade point average (GPA)** | | | | | | | |

*(Continued)*

| | | | | | | | <0.001 |
|---|---|---|---|---|---|---|---|
| GPA = 0.1–1.0 | 0.0% ± 0.0% | 0.0% ± 0.0% | 0.0% ± 0.0% | 0.0% ± 0.0% | 3.3% ± 3.4% | 0.0% ± 0.0% | |
| GPA = 1.1–2.0 | 1.1% ± 0.3% | 0.5% ± 0.3% | 0.3% ± 0.2% | 0.0% ± 0.0% | 4.7% ± 1.0% | 2.3% ± 0.5% | |
| GPA = 2.1–3.0 | 1.6% ± 0.2% | 1.6% ± 0.2% | 0.2% ± 0.1% | 0.1% ± 0.0% | 1.9% ± 0.3% | 4.4% ± 0.6% | |
| GPA = 3.1–4.0 | 1.7% ± 0.2% | 2.7% ± 0.4% | 0.3% ± 0.1% | 0.3% ± 0.1% | 1.9% ± 0.2% | 7.4% ± 0.5% | |
| Unknown | 1.4% ± 0.3% | 2.3% ± 0.4% | 0.5% ± 0.2% | 0.3% ± 0.1% | 2.4% ± 0.5% | 5.8% ± 1.0% | |

**Abbreviations:** SE = standard errors; TGD = Transgender and gender diverse; AMAB = assigned male at birth; AFAB = assigned female at birth.

*Chi-square test of association with Rao & Scott adjustment for categorical data, one-way ANOVA with adjustment for complex survey design for continuous data.

**Living with others (relatives, temple, rented house) excluded from analyses due to extremely small number.

participants who provided complete vs. incomplete information regarding gender and sexuality had different distributions of geographic region, religion, and living situation. However, these participants were not significantly different with regards to type of school attended, year level of study, weekly allowance received, and grade point average (GPA).

## Instrumentation: Behavioral health

For additional detail, we have provided a translated version of parts of the study questionnaire relevant to the analyses in this study in the supplementary material section.

*Depressive symptoms*: We measured depressive symptoms at the time of study using the Thai version of the PHQ-2 questionnaire, with the cut-off score of 3 or higher out of 6 points for having depressive symptoms at the time of study. We also measured past-year depressive symptoms with a binary question on whether the participant had a history of feeling sad or despaired on a near-daily basis for two weeks or longer within the prior 12 months. Non-responses were treated as missing values.

*Suicidality*: We asked participants whether they had seriously considered killing themselves, planned own suicide, or attempted suicide within the past 12 months. All answers were binary yes/no responses, although participants who answered about suicide attempts were also asked to specify the number of attempts made. Non-responses were treated as missing values.

*Sexual activity*: We measured lifetime history of sexual activity using the question "*Have you ever had sex*? *(Not including manual, oral, or object-based contacts)*". We limited the analyses of data related to sexual behaviors other than lifetime history (i.e., use of alcohol during last sexual encounter, use of illicit drug during last sexual encounter, foregoing contraceptive use during last sexual encounter, and condom use during last sexual encounter) to only participants who reported a lifetime history of sexual intercourse. Our methods for categorization of participants were the same as a previous study [16]. Non-responses were treated as missing values.

*Drinking, tobacco, and drug use*: Our methods of measurement in this survey were similar to the methods previously described in the literature [16]. We classified participants as ever drinkers, former drinkers, and current drinkers, based on whether they reported history of drinking within their lifetime and within the past 12 months, and used similar questions and categorization for tobacco use. For lifetime history of illicit drug use, we included only drugs where the lifetime history of use was higher than one percent in the entire population: 1) *kratom* (*Mitragyna speciosa*); 2) marijuana; 3) opium; 4) ecstasy / "love drug"; 5) ketamine; 6) heroin; 7) *yaba* (amphetamine pills), and; 8) crystal methamphetamine ("ice"). Prevalence of use of inhalants was also higher than one percent, but excluded from the analyses as the substance could be obtained over-the-counter from hardware stores. The one percent cut-point was arbitrarily chosen in order to achieve adequate statistical power to make comparisons between cisgender and transgender youths. Non-responses were treated as missing values.

## Instrumentation: Measurement of experience of violence

We used the same questions as in a previous survey to measure experience of violence within the past year [16]. We asked participants to self-report whether in the past 12 months they had experienced: 1) Physical or verbal violence victimization (with involvement of a weapon); 2) altercation with others (with injuries requiring medical treatment); 3) intimate partner violence, and; 4) sexual violence. Physical or verbal violence victimization was measured with the question "*How often did you have the following behavior or activity*?. . .*5. Being threatened or assaulted with a weapon e.g., knife, gun, bat, or other weapons.*" with two answer choices under the "*Within previous 12 months*" column: "□ *Never*" and "□ *Ever*". Altercation with others with

injuries requiring medical treatment was measured with the question "*How often did you have the following behavior or activity?. . .7. Punching / slapping / fighting with others to the point where you were injured and required medical treatment from a doctor or a nurse.*" with two answer choices under the "*Within previous 12 months*" column: "□ *Never*" and "□ *Ever*". Investigators measured the experience of intimate partner violence with the following question and answer choices: "*In the past 12 months, has your romantic partner ("แฟน") ever intentionally hit or physically assaulted you? □0) Never; □1) Yes; □2) Never had a romantic partner*". Investigators measured the experience of sexual violence with the following question and answer choices: "*In the past 12 months, have you been forced to have sex against your will? □0) Never been forced; □1) Yes; □2) Never had sex*". For these questions, we considered participants who answered "Never had a romantic partner" and "Never had sex" as those who had never experienced intimate partner violence and sexual violence, respectively. For past-year experience of violence (any type), we treated non-responses regarding each type of violence as missing values. Thus, we only included participants who answered all four questions regarding past-year experience of violence in our analyses.

## Statistical analysis

We analyzed and presented the study data using descriptive statistics with cross-tabulations. Owing to the multiple categories of gender identities in this study, we decided to present each cross-tabulations in two parts in order to accommodate to the page margins, and the results of the chi-square test with Rao-Scott adjustment were placed in the second part of the cross-tabulation tables. We compared continuous data (i.e., mean amount of weekly allowance reported by participants in each group in *Table 2*) using one-way ANOVA. Furthermore, in order to assess the extent that incomplete information regarding gender and sexuality occurred at random or the otherwise, we also compared the characteristics of participants who reported complete vs. incomplete information as supplementary material. All analyses were adjusted for sampling weight and complex survey design using the *Survey* package in R [19]. We presented prevalence data as weighed percentage ± standard error (SE), the latter of which can be regarded as a margin of potential sampling error.

## Results

There were 24,143 participants who placed their questionnaires in the envelope, among whom 23,659 (98.0%) were deemed to have filled the questionnaires adequately and were included in our analyses. Among participants included in our analyses, 32 percent identified as cisgender-heterosexual boys, 35 percent identified as cisgender-heterosexual girls, 23 percent identified as LGBTQA+ with the most common identities being bisexual/polysexual girls (7 percent) and otherwise queer and questioning, assigned female at birth (AFAB) persons (6 percent). The remaining 10 percent provided incomplete information (did not answer all three questions on sex, gender identity, and sexual orientation) (*Table 1*).

With regards to mental health outcomes and health behaviors, depressive symptoms at time of study and within past year was higher among bisexual, transgender, asexual, and otherwise queer and questioning youths compared to those who identified as cisgender-heterosexual or cisgender-homosexual, and such prevalence was generally higher among assigned female at birth participants than among their assigned male at birth counterparts (*Table 3A and 3B*). Similar general patterns were found with regard to past-year suicidality. Lifetime history of sexual activity was highest among assigned male at birth transgender persons and cisgender-homosexual boys, and lowest among those who identified as asexual. Participants who identified as LGBTQA+ generally had higher prevalence of lifetime alcohol use than

**Table 3.** a. Prevalence (weighted percent ± SE) of mental health outcomes and health behaviors among participants, part 1. b. Prevalence (weighted percent ± SE) of mental health outcomes and health behaviors among participants, part 2.

| | Cisgender Heterosexual Boys (n = 7,810) | Cisgender Heterosexual Girls (n = 8,225) | Cisgender Homosexual Boys (n = 496) | Cisgender Homosexual Girls (n = 619) | Cisgender Bisexual / Polysexual Boys (n = 199) | Cisgender Bisexual / Polysexual Girls (n = 1,457) |
|---|---|---|---|---|---|---|
| **_Depressive symptoms_** | | | | | | |
| PHQ-2 score at time of study (mean ± standard errors) | | | | | | |
| Depressive symptoms at time of study | 8.6%±0.6% | 14.5%±0.8% | 9.2%± 2.4% | 14.5%± 2.5% | 21.0%± 4.4% | 26.8%± 1.7% |
| Depressive symptoms in past 12 months | 14.2%± 0.8% | 22.2%±1.0% | 13.9%± 2.4% | 21.1%± 2.7% | 25.0%± 3.9% | 37.0%± 1.9% |
| **_Suicidality in past 12 months_** | | | | | | |
| Suicidal ideation | 4.6%±0.5% | 10.7%±0.8% | 5.5%± 1.5% | 9.8%± 1.9% | 13.7%± 3.7% | 20.6%± 1.6% |
| Suicide planning | 4.8%±0.5% | 11.4%±0.8% | 6.8%± 1.6% | 8.7%± 2.0% | 12.8%± 3.1% | 23.5%± 1.4% |
| Suicide attempt | 3.4%±0.3% | 7.9%±0.7% | 3.4%± 1.1% | 7.1%± 1.7% | 6.6%±2.3% | 15.3%± 1.3% |
| **_Sexual Behaviors_** | | | | | | |
| Ever had sex | 7.2%±1.0% | 5.3%±0.7% | 15.5%± 3.9% | 7.1%± 0.4% | 12.7%± 2.7% | 3.6%±0.8% |
| _Among those who ever had sex_ | | | | | | |
| Use of alcohol during last sexual encounter | 17.5%±2.3% | 13.8%±2.4% | 11.0%± 4.8% | 17.6%± 9.1% | 3.2%±2.6% | 18.3%± 7.7% |
| Use of illicit drug during last sexual encounter | 6.1%±1.2% | 1.4%±0.6% | 14.8%± 5.5% | 0.0%± 0.0% | 5.3%±4.1% | 0.9%±0.9% |
| Foregoing contraceptive use during last sexual encounter | 18.5%±1.7% | 10.9%±3.2% | 11.0%± 5.1% | 19.3%± 8.1% | 24.8%± 12.2% | 23.3%± 12.4% |
| Condom use during last sexual encounter | 65.4%±2.8% | 71.7%±2.8% | 72.9%± 6.9% | 64.5%± 10.9% | 67.8%± 11.8% | 67.7%± 11.6% |
| **_Alcohol and tobacco use_** | | | | | | |
| Ever drinker | 25.6%±2.1% | 27.9%±2.1% | 30.7%± 5.5% | 33.3%± 3.9% | 28.6%± 4.0% | 32.6%± 3.0% |
| Drank in past 12 months (among ever drinkers) | 74.4%±1.7% | 81.6%±1.3% | 81.2%± 5.4% | 74.5%± 3.7% | 68.4%± 7.1% | 75.2%±2.5 |
| Ever smoker | 6.2%±0.7% | 1.3%±0.2% | 9.3%± 3.0% | 3.2%± 1.2% | 3.1%±1.6% | 2.4%±0.4% |
| Smoked in past 12 months (among ever smokers) | 60.2%±2.4% | 55.1%±2.6% | 74.0%± 6.3% | 49.1%± 8.7% | 47.3%± 12.4% | 65.9%± 8.0% |
| **_Lifetime history of illicit drug use_** | | | | | | |
| Kratom** | 7.2%±1.0% | 2.7%±0.5% | 8.7%± 2.2% | 2.7%± 0.7% | 6.8%±1.7% | 3.4%±0.6% |
| Marijuana | 5.6%±0.6% | 1.3%±0.2% | 7.5%± 1.9% | 2.6%± 0.8% | 5.4%±2.0% | 2.4%±0.6% |
| Opium | 1.4%±0.2% | 0.4%±0.1% | 3.9%± 1.3% | 1.6%± 0.6% | 4.3%±1.9% | 0.4%±0.2% |

_(Continued)_

**Table 3.** (Continued)

| | TGD, AMAB (n = 342) | TGD, AFAB (n = 395) | Asexual, AMAB (n = 60) | Asexual, AFAB (n = 50) | Otherwise queer and questioning, AMAB (n = 433) | Otherwise queer and questioning, AFAB (n = 1,237) | P-value* |
|---|---|---|---|---|---|---|---|
| Ecstasy / Love drug | 1.4%±0.2% | 0.5%±0.1% | 0.9%±0.4% | 3.6%±1.2% | 4.0%±1.7% | 0.4%±0.2% | |
| Ketamine | 1.5%±0.2% | 0.8%±0.2% | 1.3%±0.5% | 4.3%±1.3% | 4.2%±1.7% | 0.4%±0.2% | |
| Heroin | 1.4%±0.2% | 0.5%±0.1% | 1.1%±0.5% | 4.4%±1.4% | 2.9%±1.3% | 0.3%±0.2% | |
| Inhalants | 1.4%±0.2% | 0.5%±0.1% | 0.9%±0.4% | 3.4%±1.2% | 4.7%±1.8% | 0.5%±0.2% | |
| Yaba (methamphetamine pills) | 1.7%±0.3% | 0.5%±0.1% | 0.9%±0.4% | 3.8%±1.3% | 2.6%±1.3% | 0.4%±0.2% | |
| Ice (crystal methamphetamine) | 1.6%±0.3% | 0.6%±0.1% | 0.9%±0.4% | 4.2%±1.3% | 3.9%±1.7% | 0.2%±0.2% | |
| *Depressive experience* | | | | | | | |
| PHQ-2 score at time of study (mean ± standard errors) | | | | | | | |
| Depressive symptoms at time of study | 18.8%±2.1% | 39.8%±3.5% | 13.6%±5.1% | 20.9%±7.2% | 18.5%±2.8% | 29.0%±1.7% | **<0.001** |
| Depressive symptoms in past 12 months | 28.9%±2.8% | 47.4%±2.7% | 11.6%±4.9% | 34.3%±10.2% | 23.4%±2.8% | 36.6%±2.0% | **<0.001** |
| *Suicidality in past 12 months* | | | | | | | |
| Suicidal ideation | 17.1%±2.2% | 35.4%±3.5% | 4.9%±2.5% | 18.7%±7.5% | 12.7%±1.7% | 23.8%±1.7% | **<0.001** |
| Suicide planning | 18.7%±2.9% | 36.4%±2.2% | 15.4%±8.8% | 16.7%±6.8% | 11.5%±2.3% | 26.7%±1.5% | **<0.001** |
| Suicide attempt | 13.9%±2.5% | 19.4%±1.6% | 10.2%±4.7% | 5.3%±3.3% | 8.7%±1.7% | 16.9%±1.7% | **<0.001** |
| *Sexual Behaviors* | | | | | | | |
| Ever had sex | 18.1%±2.5% | 6.6%±1.6% | 2.2%±1.7% | 0.0%±0.0% | 7.5%±1.9% | 3.2%±0.6% | **<0.001** |
| *Among those who ever had sex* | | | | | | | |
| Use of alcohol during last sexual encounter | 19.5%±7.2% | 2.9%±2.9% | 46.8%±35.5% | 0.0%±0.0% | 2.0%±6.9% | 16.5%±9.2% | 0.641 |
| Use of illicit drug during last sexual encounter | 5.4%±3.2% | 2.9%±3.0% | 46.8%±35.5% | 0.0%±0.0% | 17.7%±6.8% | 7.2%±6.8% | **0.003** |
| Foregoing contraceptive use during last sexual encounter | 28.7%±6.8% | 25.4%±9.6% | 53.2%±35.5% | 0.0%±0.0% | 15.8%±7.5% | 21.2%±7.2% | 0.317 |
| Condom use during last sexual encounter | 60.7%±8.2% | 49.3%±12.7% | 0.0%±0.0% | 0.0%±0.0% | 58.5%±10.6% | 55.3%±10.6% | 0.510 |
| *Alcohol and tobacco use* | | | | | | | |
| Ever drinker | 39.2%±3.3% | 40.0%±2.5% | 13.1%±5.5% | 8.8%±4.4% | 29.1%±2.8% | 32.6%±2.7% | **<0.001** |

*(Continued)*

**Table 3.** (Continued)

| | | | | | | | p-value |
|---|---|---|---|---|---|---|---|
| **Drank in past 12 months (among ever drinkers)** | 72.6%±4.7% | 78.5%±3.3% | 59.3%±23.5% | 86.9%±11.2% | 68.9%±65.0% | 77.0%±2.1% | **0.010** |
| **Ever smoker** | 3.3%±1.1% | 3.3%±1.0% | 1.3%±1.1% | 0.0%±0.0% | 4.4%±1.3% | 0.8%±0.3% | **<0.001** |
| **Smoked in past 12 months (among ever smokers)** | 75.1%±8.9% | 56.5%±8.9% | 68.4%±30.8% | 0.0%±0.0% | 55.1%±8.4% | 51.9%±7.7% | 0.335 |
| *Lifetime history of illicit drug use* | | | | | | | |
| **Kratom** | 4.0%±1.6% | 4.0%±1.3% | 1.8%±1.3% | 2.2%±2.2% | 4.9%±1.6% | 2.0%±0.6% | **<0.001** |
| **Marijuana** | 3.3%±1.6% | 3.5%±1.9% | 1.8%±1.3% | 0.0%±0.0% | 4.8%±1.6% | 1.3%±0.4% | **<0.001** |
| **Opium** | 1.3%±0.6% | 0.5%±0.4% | 1.4%±1.2% | 0.0%±0.0% | 1.6%±0.7% | 0.6%±0.2% | **<0.001** |
| **Ecstasy / Love drug** | 1.3%±0.7% | 0.7%±0.6% | 1.4%±1.2% | 0.0%±0.0% | 1.7%±0.9% | 0.6%±0.2% | **<0.001** |
| **Ketamine** | 1.3%±0.7% | 0.5%±0.4% | 1.4%±1.2% | 0.0%±0.0% | 1.3%±0.6% | 0.6%±0.2% | **<0.001** |
| **Heroin** | 1.3%±0.7% | 0.5%±0.4% | 1.4%±1.2% | 0.0%±0.0% | 1.4%±0.8% | 0.6%±0.2% | **<0.001** |
| **Yaba (methamphetamine pills)** | 1.2%±0.7% | 1.2%±0.7% | 1.4%±1.2% | 0.0%±0.0% | 1.3%±0.6% | 0.6%±0.2% | **<0.001** |
| **Ice (crystal methamphetamine)** | 1.8% ± 0.8% | 1.1% ± 0.6% | 1.4% ± 1.2% | 0.0% ± 0.0% | 1.5% ± 0.7% | 0.4% ± 0.2% | **<0.001** |

**Abbreviations:** SE = standard errors; TGD = Transgender and gender diverse; AMAB = assigned male at birth; AFAB = assigned female at birth.

*Chi-square test of association with Rao & Scott adjustment. Bold p-values denote statistically significant association.

**Includes both kratom and kratom mixture (4 x 100).

cisgender-heterosexual participants, whereas those who identified as asexual had the lowest prevalence of alcohol use. Participants who were assigned male at birth generally had higher prevalence of alcohol use than their assigned female at birth counterparts. Similar patterns were also observed for lifetime history of illicit drug use.

History of past-year experiences of violence varied widely between study groups (*Table 4A and 4B*). In general, participants who identified as asexual reported the lowest prevalence of all types of violence. Meanwhile, participants who identified as homosexuals, bisexual or polysexual boys, and otherwise queer and questioning assigned male at birth reported the highest prevalence.

## Discussion

In a nationally-representative survey, we identified secondary students in Thailand by sex assigned at birth, gender identity, and sexual orientation and reported on differences in behavioral health and experience of violence between groups. We found differences between LGBTQA+ and cisgender-heterosexual participants with regards to prevalence of depressive symptoms, suicidality, lifetime sexual activity, alcohol and tobacco use, lifetime history of illicit drug use, and past-year exposure to violence. We hope that the findings of our analyses can serve as useful basic information for stakeholders in adolescent health and LGBTQA+ issues.

Approximately 23.2% of our study participants identified as LGBTQA+ whereas another 9.6% of the participants did not answer all three questions required for proper categorization and were excluded from our main analyses. Supplementary findings showed that participants who were excluded had similar characteristics to those who did. This suggested that that information bias from non-responses was unlikely to affect the study findings. A cautionary interpretation, however, would be that the estimated prevalence of LGBTQA+ identity was anywhere between 23.2% to 32.8% of all participants. We found that participants in Matthayom 3 and Matthayom 5 were more likely to identify as LGBTQA+ than participants in Matthayom 1 and those in vocational education. These differences could be attributed to the process of self-identification occurring more commonly at a later age [20]. On the other hand, a more hostile institutional environment [21] could have induced social desirability bias among some participants.

The prevalence of LGBTQA+ identity in our study was higher than a previously-estimated 8 percent prevalence among Thais [22] and the global average of 11 percent [23]. However, such difference might reflect the cohort effect among the participants [23,24]. Nearly all of our participants were born between 2003 and 2008 at the time of study, and could be considered as belonging to Generation-Z, which more commonly identify as LGBTQA+ than previous generations [25]. The less-stigmatized social contexts (compared to previous generations) might have enabled higher level of disclosure compared to previous generations of LGBTQA+ youths. The prevalence of those who identified as transgender or gender diverse in our study was lower than another study among secondary school students in Thailand who found prevalence of gender non-conforming identity at 9.1% [26], although such categorization also could have included those who were otherwise queer and questioning, making comparison difficult. An additional caveat is that our younger participants were in their early adolescence and may be exploring their sense of identity. Our cross-sectional study data only presented the identity at the time of study, and our study design precluded the measurement of these potential shifts.

We attempted to improve our gender identity question in this round of the survey by changing the question wording to more closely reflect the notion of identity [16]. We also included the answer choices of "gender diverse" and "not sure". However, the term "gender diverse" (Thai: *Phet thang luek*) was also used as an umbrella term to refer to the LGBTQ

**Table 4. a.** Prevalence (weighted percent ± SE) of past-year exposure to violence among the participants, part 1. **b.** Prevalence (weighted percent ± SE) of past-year exposure to violence among the participants, part 2.

| | Cisgender Heterosexual Boys | Cisgender Heterosexual Girls | Cisgender Homosexual Boys | Cisgender Homosexual Girls | Cisgender Bisexual / Polysexual Boys | Cisgender Bisexual / Polysexual Girls |
|---|---|---|---|---|---|---|
| | (n = 7,810) | (n = 8,225) | (n = 496) | (n = 619) | (n = 199) | (n = 1,457) |
| **Past-year exposure to violence** | | | | | | |
| Physical or verbal violence victimization (with involvement of a weapon) | 5.0%±0.5% | 1.7%±0.2% | 8.1%±2.9% | 2.6%±1.1% | 7.1%±1.7% | 2.2%±0.6% |
| Altercation with others (with injuries requiring medical treatment) | 5.7%±0.6% | 1.7%±0.2% | 6.2%±2.0% | 3.3%±1.0% | 5.6%±1.8% | 2.6%±0.6% |
| Intimate partner violence | 3.8%±0.5% | 1.5%±0.3% | 7.3%±2.0% | 3.3%±1.2% | 4.2%±1.9% | 1.7%±0.4% |
| Sexual violence | 1.3%±0.2% | 1.2%±0.2% | 5.2%±1.4% | 1.6%±0.7% | 1.4%±0.8% | 1.1%±0.3% |
| Experienced any type of violence in past 12 months | 10.8%±0.9% | 4.4%±0.3% | 15.6%±2.7% | 7.2%±1.4% | 10.6%±2.1% | 5.1%±0.9% |

| | TGD, AMAB | TGD, AFAB | Asexual, AMAB | Asexual, AFAB | Otherwise queer and questioning, AMAB | Otherwise queer and questioning, AFAB | p-value* |
|---|---|---|---|---|---|---|---|
| | (n = 342) | (n = 395) | (n = 60) | (n = 50) | (n = 433) | (n = 1,237) | |
| **Past-year exposure to violence** | | | | | | | |
| Physical or verbal violence victimization (with involvement of a weapon) | 2.7%±1.0% | 3.7%±1.3% | 2.2%±1.3% | 0.0%±0.0% | 8.6%±2.3% | 2.1%±0.5% | **<0.001** |
| Altercation with others (with injuries requiring medical treatment) | 4.3%±1.2% | 1.7%±0.7% | 3.9%±2.2% | 0.0%±0.0% | 8.2%±1.9% | 2.4%±0.6% | **<0.001** |
| Intimate partner violence | 2.1%±0.8% | 4.4%±1.6% | 1.5%±1.1% | 0.0%±0.0% | 3.9%±1.2% | 0.8%±0.3% | **<0.001** |
| Sexual violence | 1.7%±0.7% | 2.5%±0.7% | 0.0%±0.0% | 0.0%±0.0% | 1.6%±0.8% | 1.4%±0.3% | **0.003** |
| Experienced any type of violence in past 12 months | 7.3%±1.6% | 9.4%±1.8% | 4.5%±2.4% | 0.0%±0.0% | 14.4%±2.8% | 4.7%±0.8% | **<0.001** |

**Abbreviations:** SE = standard errors; TGD = Transgender and gender diverse; AMAB = assigned male at birth; AFAB = assigned female at birth.

*Chi-square test of association with Rao & Scott adjustment. Bold p-values denote statistically significant association.

community [27,28]. Similarly, our sexual orientation measurement question ("*H1b. To which gender are you attracted*?") did not distinguish between romantic attraction and sexual attraction. Thus, the labels of "heterosexual", "homosexual", "bisexual/polysexual", and "asexual" in our study could refer to either romantic or sexual orientation, depending on the perception of the individual participant [2]. In that regard, participants with "mixed orientation" or "cross-orientation" could be misclassified with regard to either their romantic or sexual orientation [3]. These issues should be considered when interpreting the findings of this study.

Compared to adult Thai LGBTQA+ population [29], our participants generally had lower prevalence of past-year suicidal ideation and suicide attempt. However, there were notable variations between groups. Concerningly, those who identified as assigned female at birth transgender having the highest prevalence. Meyer's Minority Stress model offers a theoretical framework that stigma, prejudice, and discrimination faced by LGBTQA+ people "create a hostile and stressful social environment that causes mental health problems" [30], albeit the model may need to be further modified [31]. A survey in Australia among young people attracted to those of the same sex showed that internalized homophobia, perceived stigma, and experiences of homophobic physical abuse were associated with suicidal thoughts [32]. It is possible that adolescent LGBTQA+ individuals in our study were also subject to similar stressors, such as sexual/gender stigma [29] and internalized discrimination [33], which then influenced their suicidality.

Our lifetime history of sexual activity question in this round of survey did specify that the definition of sex did not include manual, oral, or object-based activities, which helped to reduce the issue of ambiguity on what is considered as "sex" [34]. In that regard, the high prevalence of high-risk sex among sexually-active assigned male at birth asexual participants should be interpreted only in the context of estimation based on a very limited number of samples with high standard errors of the estimates, i.e., high level of potential errors.

The prevalence of lifetime drinking and current drinking varied widely among our study participants, although those who identified as transgender had relatively high prevalence and those who identified as asexual had relatively low prevalence compared to all other groups, which differed from the findings of a previous round of survey [16]. However, the past-year drinking history in this analysis coincided with the first year of the COVID-19 pandemic, during which LGBTQ youths were disproportionately affected with regard to mental health [35]. The higher prevalence of drinking in certain groups could be a reflection of unhealthy coping mechanisms [36], or a reflection of broader societal trends in which alcohol consumption is declining among adolescents and youths [37]. Such disparities were also observed for lifetime history of illicit drug use. Homosexual male participants seemed to have notably higher prevalence in nearly all types of substances compared to other groups. This differed from the findings of a national survey on drug use in the United States, which found that bisexual women and bisexual men (in addition to gay men) had 2–3 times higher prevalence of substance use compared to heterosexual adults [38]. These lack of disparities in LGBTQA+ groups other than gay males should be further investigated. In that regard, although kratom and cannabis were classified as Category V narcotics when we collected data in this study, Thailand recently legalized these substances [39]. Thus, the findings of this study should be generalized only to the pre-legalization context.

Past-year experience of violence varied widely among our participants, but those who were assigned male at birth generally experienced violence more than those who were assigned female at birth with the same gender and sexual orientations. The findings of this study further expand the understanding gained from a previous study [26], which found that social violence negatively correlated with the extent of conformity with sex assigned at birth. In our study, participants who were gay, lesbian, transgender assigned female at birth, and queer-and-

questioning assigned male at birth were more likely to experience violence (any type) within the past year. These findings suggested that social violence among Thai students may also vary by sexual orientation in addition to gender conformity. One potentially problematic issue with our study findings was the classification of participants who answered that they never had a romantic partner or never had sex as those who never experienced intimate partner violence or sexual violence, respectively. This classification could be misleading. The definitions of having a romantic partner or having had sex were not included as part of the question, which could have introduced misclassification error in the responses, leading to potential information bias. The questions in our study and the analyses methods eased comparison of the findings with a previous study [16], but should not preclude improvement in future studies. The act of being threatened (i.e., victimization of verbal violence or simple assault) was in the same item as being actually assaulted with a weapon (i.e., victimization of physical violence or aggravated assault) (i.e., Question I5). These acts should be measured in separate questions in future studies. Similarly, questions regarding physical altercations (i.e., Questions I6 and I7) did not distinguish between perpetration (either as an instigator or as an act of self-defense or retaliation) and victimization. Future studies should consider making such distinctions clear and measure instigation, self-defense, retaliation, and victimization of violence separately.

## Strengths and limitations

The strengths of our study were the large sample size and the method of classification of participants, which allowed for assessment of health disparities between LGBTQA+ and cisgender-heterosexual youths with considerable statistical power. However, a number of limitations should be considered in the interpretation of our study findings. Firstly, we classified participants who did not fit the rigid categorization system as "otherwise queer and questioning", which precluded us from showing the full range of diversity of human identities. Nonetheless, we attempted to keep our categorization coherent with the existing conceptualization of LGBTQA+ identities. Secondly, our study data was collected in early 2021, thus the past-year history of health behaviors in our study data may be generalizable only in the context of the COVID-19 pandemic. Question to measure depressive experience within the past year explicitly referred to having the symptoms for 2 consecutive weeks or longer instead of the frequency (number of days) in which symptoms according to the PHQ appeared in a 2-weeks period. Those who had non-daily depressive symptoms, albeit with considerable frequency, could have been misclassified as not having depressive symptoms. This outcome misclassification should be considered as a potential source of information bias. Thirdly, our study did not include youths outside the formal education system, further limiting the generalizability of our study findings.

## Suggestions for future studies

With regard to measurement of identities, specifically in the Thai context, future studies should consider improving inclusivity by including the terms "genderfluid" and "nonbinary" in addition to the existing answer choices, using the current version of these terms in the Thai language [40,41]. In addition, future studies should consider separating the question of romantic attraction from the question of sexual attraction in order to capture the diversity of human romantic and sexual orientations. Future studies should also consider the use of non-traditional labels of gender identities and romantic or sexual orientations in addition to traditional ones [42].

Considering Thailand's recent decriminalization of kratom and cannabis, future studies should consider measuring kratom and cannabis use in more details as separate sections on

the questionnaire by adapting existing instruments [43–45]. Furthermore, potential misclassification from not presuming that those who never had a romantic partner or never had sex had never experienced intimate partner violence or sexual violence can be reduced in future studies by providing or reiterating, in a clear manner, the definitions of having a romantic partner and having had sex.

## Conclusion

We provided a national-level estimates of proportion of secondary students in Thailand who identified as LGBTQA+, and found disparities in behavioral health between cisgender-heterosexual and LGBTQA+ participants. This is one of the first studies to make assessments on such scales and to include such diverse number of identities. However, issues regarding potential misclassification of participants, limitation of past-year history of behaviors to the context of the COVID-19 pandemic, and the lack of data from youths outside the formal education system should be considered as caveats in the interpretation of the study findings.

## Supporting information

**S1 File. Anonymized data set.** Anonymized data set to replicate the study findings.
(CSV)

**S2 File. R Codes.** Codes for data analyses, text file with annotations.
(TXT)

**S3 File. STROBE checklist.** STROBE checklist for cross-sectional studies.
(DOCX)

**S4 File. Questionnaire.** Partial English-language translation of the study questionnaire.
(DOCX)

**S5 File. Supplementary Table 1.** Characteristics of study participants who provided complete vs. incomplete information regarding gender and sexuality.
(DOCX)

## Acknowledgments

We wish to thank all study participants for their time and energy in completing our study questionnaire. We also wish to thank all regional data collection team, and the data entry and data management staff, for their tireless efforts in making this study possible.

## Author Contributions

**Conceptualization:** Wit Wichaidit, Natnita Mattawanon, Virasakdi Chongsuvivatwong, Sawitri Assanangkornchai.

**Data curation:** Wit Wichaidit.

**Formal analysis:** Wit Wichaidit, Witchaya Somboonmark, Nattaphorn Prodtongsom.

**Funding acquisition:** Sawitri Assanangkornchai.

**Investigation:** Wit Wichaidit.

**Methodology:** Wit Wichaidit.

**Project administration:** Sawitri Assanangkornchai.

**Resources:** Sawitri Assanangkornchai.

**Software:** Virasakdi Chongsuvivatwong.

**Supervision:** Sawitri Assanangkornchai.

**Validation:** Wit Wichaidit, Natnita Mattawanon, Witchaya Somboonmark, Nattaphorn Prodtongsom, Virasakdi Chongsuvivatwong.

**Writing – original draft:** Wit Wichaidit.

**Writing – review & editing:** Wit Wichaidit, Natnita Mattawanon, Virasakdi Chongsuvivatwong.

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
