## [Decision Letter · Decision Letter 0]

25 Jan 2023

PONE-D-22-31754Disparities in Behavioral Health and Experience of Violence between Cisgender-Heterosexual vs. Lesbian, Gay, Bisexual, Transgender, Queer and Questioning, and Asexual (LGBTQA+) Thai AdolescentsPLOS ONE

Dear Dr. Wichaidit,

Thank you for submitting your manuscript to PLOS ONE. After careful consideration, we feel that it has merit but does not fully meet PLOS ONE’s publication criteria as it currently stands. Therefore, we invite you to submit a revised version of the manuscript that carefully addresses reviewer's suggestions.

We look forward to receiving your revised manuscript.

Kind regards,

Marianna Mazza

Academic Editor

PLOS ONE

Journal Requirements:

2. In the ethics statement in the Methods, you have specified that verbal consent was obtained. Please provide additional details regarding how this consent was documented and witnessed, and state whether this was approved by the IRB

“The funders had no role in study design, data collection and analysis, decision to publish, or preparation of the manuscript”

Reviewers' comments:

Reviewer's Responses to Questions

**Comments to the Author**

1. Is the manuscript technically sound, and do the data support the conclusions?

Reviewer #1: Yes

Reviewer #2: Partly

Reviewer #3: Partly

2. Has the statistical analysis been performed appropriately and rigorously? 

Reviewer #1: Yes

Reviewer #2: No

Reviewer #3: No

3. Have the authors made all data underlying the findings in their manuscript fully available?

Reviewer #1: Yes

Reviewer #2: Yes

Reviewer #3: Yes

4. Is the manuscript presented in an intelligible fashion and written in standard English?

Reviewer #1: Yes

Reviewer #2: Yes

Reviewer #3: Yes

5. Review Comments to the Author

Reviewer #1: The manuscript brings relevant information about the literature and clearly supports its justification and aims. Also, the study is relevant for publication and highlights important information about the health-disease process of LGBTQA youth and also their health disparities. The paper approaches a relevant topic, the authors use appropriate weighting procedures to overcome the complex design and presented data from a large national school survey. Nonetheless, despite its relevance, design and appropriate conclusions, some minor issues remain. These issues are described in the attached document.

Reviewer #2: I was enthusiastic about reading this paper on sexual attraction and gender identity-based differences in health among adolescents in Thailand. However, I think the paper can be approved on. There is not a lot of rationale as to why to study LGBTQ+ adolescents. I am also not convinced by the operationalization of the LGBTQ+ groups. Because of this, I am not sure how valuable these findings really are or how they should be interpreted. They authors should come up with a strong rationale for this operationalization or change it. See below my comments.

Abstract

1. The term “cisgender-heteronormative persons” reads awkward to me. I believe it is more common to use a term like “heterosexual or cisgender people”. I would suggest to change this throughout.

2. It is not clear from the background section that the study focuses on adolescents. Some rationale is needed why studying sexual and gender identity-based health disparities specifically among adolescents is needed.

3. In the methods or results section, it would be helpful to also report the mean age and standard deviation of the sample.

4. Violence is mentioned throughout the abstract, but I am not sure what is meant by this. Is this physical violence, verbal violence? Is it akin to discrimination?

5. In the first sentence of the “Results” section of the abstract, the number is 23,659 is mentioned twice, which is unnecessary

6. It is first stated that 23,659 participants returned complete and valid questionnaires. Then it stated that “10 percent provided incomplete information with regard to gender identity and sexual orientation”. I do not understand this.

Introduction

7. I am not sure whether I agree with the definition following definition: “the term "transgender" refers to the state in which an individual's gender identity is different from their sex assigned at birth.” Someone who identifies as nonbinary has a different gender identity then the assigned sex at birth, but is not considered transgender. Also, the word “state” reads awkward here. I would suggest to search for references in which more correct definitions are used. The same holds for the definition of sexual orientation, which also seems incomplete to me.

8. I do not understand the sentence “(LGBTQA+) refers to any person who does not identify as cisgender-heteronormative, regardless of their gender identity, sexual orientation, or asexuality”. If you do not identify as “cisgender-heteronomrative” than you will probably identify as LGBTQ+, so I do not understand the “regardless of their gender identity, sexual orientation, or asexuality” part of this definition. Also, when someone does identify as cisgender-heteronormative, but is somewhat attracted to people of the same gender, how would they then be categorized? The authors should put more effort in providing a clear description of what they mean with terms they are using. Right now, it seems there is only a surface level understanding of sexual identity, behavior, attraction and gender identity. These concepts overlap, but are not the same and while reading this, I do not get a sense that authors understand these differences. Asexuality is further often considered as a sexual orientation, singling is out here seems incorrect.

9. What is meant with “social and human rights issues”? Similarly, what do you mean with violence and abuse? Further, more detail is needed when discussing findings of other (empirical?) studies.

10. On line 64, your mention health disparities. Compared to whom have research identified these disparities?

11. I think a little more attention can be paid to that most research on LGBT+ youth is not conducted in Thailand. I am also not sure whether the few studies conducted in Thailand focus on adolescents. It should also be made clearer why it is crucial to study these disparities during adolescence. What precisely is the research gap?

12. Last, I am also not sure about the whole structure of the introduction. Only form the third paragraph onward it becomes clear what the goal of this study is. You should try to revise the structure of this section to make it clear from the start what you are studying, what the research question is, and who the population is.

Materials and Methods

13. When discussing the sample, make sure to mention the typical ages of the participants. I am not sure how old students in year 7, 9 and 11 are. I am also wondering why 8 and 10 were not included in the study.

14. I am also wondering how the survey was administered (paper pencil during a class?), if participation was voluntary, and whether consent was obtained?

15. The sentence “We used the responses of three questions to define the genders in this study” is awkward. More correct would be to discuss that three questions were used to assess gender identity. It is also incorrect that questions on sexual attraction are used to inform gender identity. Again, a better understanding is needed of how gender identity and sexual orientation are distinct.

16. It is irrelevant to mention from which sections of the questionnaire certain question came from.

17. When discussing questions, it would be helpful if a translated question is presented, next to the already present answer options.

18. I am also not sure about the sentence “We used the responses of three questions to define the genders in this study.” You are also mentioning sexual attraction questions here, which is different from gender identity. This sentence does not reflect this. In general, I am not sure why gender identity and sexual attraction were combined this way. What is the relevance/rationale behind this? What do we gain from using this operationalization?

19. Sexual attraction is measured in this study. However, participants are described as being heterosexual and gay. But these identities were not measured. I would be more correct to refer to your participants not by using identity labels, but by their attractions (e.g., other gender attracted, same gender attracted).

20. I also have difficulty understanding why some someone who is “Cisgender Homosexual (“Gay”) Boys” is someone who is only attracted to cis-gender boys. If someone is attracted to cisgender boys and transgender boys, they can be considered gay, because they are attracted to men. Why did the authors make this distinction?

21. Why are depressive experience and suicidality considered as behavioral health?

22. For depressive experience, can you provide a sample question and the answer categories and how one score was obtained?

23. For sexual activity, you mention that “questions were largely similar to the previous round of study”. I do not know what this refers to, as I am not familiar with the “previous round”. Is this information really needed? It is also mentioned here that sexual orientation was measured. Why was this not used to measure sexual orientation, but was sexual attraction used instead?

24. For drinking, tobacco, and drug use, please provide sample questions and answer options and describe more precisely how measures were constructed.

25. For experiences with violence, what were the answer options?

26. The “Procedure” section should be moved up to the beginning of the Materials and Methods section.

27. In the data analysis section, weekly allowance is mentioned. This wat not mentioned in the measurement section, I am not sure what this is referring to.

28. Did the authors also look into missing data mechanisms. Was the data, for instance, missing at random?

29. The “Ethical Considerations” section should be moved up to the beginning of the Materials and Methods section.

30. In general, why are only bivariate associations considered? Why are no control variables introduced to the models? This would make a more convincing paper.

Results

31. Again, not sure what you mean by complete questionaries when also incomplete questions are mentioned.

32. On page 13, control variables are mentioned that were not introduced in the matarials and method section.

33. It is mentioned that “Generally, students at government schools and those in Matthayom 3 and Matthayom 5 more commonly identified as LGBTQA+ than students at private schools and students in Matthayom 1 or vocational schools.” Why is this the case?

34. I have a lot of difficulty really understanding the results. This is mainly because the “gender identity groups” are not in line with previous research and therefore hard to interpret what these differences mean. I strongly advise to change this.

35. You should mention in the tables what the referent category is. It is hard to read the tables now.

Discussion

36. I do not think that a sentence like “We found that LGBTQA+ youths overall had higher prevalence of depression, suicidality, lifetime sexual activity, alcohol and tobacco use, lifetime history of illicit drug use, and past-year exposure to violence” should be included, as you mention directly afterwards that there is heterogeneity. By stating this, the heterogeneity is not paid attention to. In general, more attention should be given to these findings and what the implications are.

37. When discussing the high numbers of LGBTQ participants, I think that you should also refer to other studies that found high prevalence among Gen Z compared to previous generations.

38. In general, more effort should be put into referring to previous studies in understanding the current results. For instance, it is mentioned that “Teenagers who initially identify as transgender may, later in their adolescence, identified as gender diverse and decided not to undergo the transitioning process”. Does this happen often? Is there any information on this?

39. When making recommendations on gender identity questions it is recommended to use more inclusive terms. Is this in general a recommendation, or specifically for the Thai context?

Reviewer #3: Manuscript Number PONE-D-22-31754

"Disparities in Behavioral Health and Experience of Violence between Cisgender-Heterosexual vs. Lesbian, Gay, Bisexual, Transgender, Queer and Questioning, and Asexual (LGBTQA+) Thai Adolescents"

This study is based on the 5th National School Survey on Alcohol Consumption, Substance Use, and Other Health-Risk Behaviors. This study sought to examine how behavioral health outcomes and exposure to violence differed between cisgender-heterosexual youths and LGBTQA+ youths.

Major comments

Q1. Suggested authors extended the concept of 1)disparities in Behavioral Health and 2) experience of Violence between CisgenderHeterosexual comprises Lesbian, Gay, Bisexual, Transgender, Queer and Questioning, and Asexual (LGBTQA+), and 3) the reason choose Thai Adolescents

Q2. Suggested the wording in the abstract is within 250 words.

Q3. The introduction has some literature review content, so a section for literature review should be created.

Q4. Suggested to rewrite 82-147, the instrumentation section needs to be shorter and specifically highlight how the instrument is a good fit with this study.

Q5. Suggest adding the statistically analyze section.

Q6. In line 195, a) even the authors draw the results Percent ± SE*; however, what are the results under meanings behind? b) what is the comparison of the results from all groups?

Q7. Suggested clear Inclusion and exclusion criteria with the paragraph.

Q8. What are the new insights from this paper, and how would the author suggest adding a section on future implications and limitations?

Q9. When the results are presented in the result section, how do you consider the results significant to the (LGBTQA+)?

Q10. Please provide background information concerning LGBTQA+ differences in behavioral health and experiences of violence to support teens and compare them with adults, even older adults.

Q11. Suggested double check and explain the results in table 1 group 9 and 10 0.3%±0.0% and 0.2%±0.0%, the ±0.0% what is the implication?

Q12. Family house/flat (n=18,461), p-value 0.115, what is the meaning of p-value when it refers to the mental health and violence experience of the teenage LGBTQA+ group?

Q13. The revised manuscript suggested citing the relevant reference in the following papers.

doi: 10.3389/fpsyg.2021.677734; doi:15579883221120985.; doi: 10.3389/fpsyg.2022.726343; doi: 10.3389/fpsyg.2021.704995; doi: 10.3389/fpsyg.2021.692343

The suggested author submits the manuscript to the editing service to ensure the manuscript meets the requirement of language quality.

6. PLOS authors have the option to publish the peer review history of their article (what does this mean?). If published, this will include your full peer review and any attached files.

Reviewer #1: No

Reviewer #2: No

Reviewer #3: No

---

## [Author Response · Author response to Decision Letter 0]

8 May 2023

Review Comments to the Author

################################################ 

Reviewer #1 

REVIEWER’S COMMENT:

The manuscript brings relevant information about the literature and clearly supports its justification and aims. Also, the study is relevant for publication and highlights important information about the health-disease process of LGBTQA youth and also their health disparities. The paper approaches a relevant topic, the authors use appropriate weighting procedures to overcome the complex design and presented data from a large national school survey. Nonetheless, despite its relevance, design and appropriate conclusions, some minor issues remain. These issues are described in the attached document. 

RESPONSE: 

We thank the reviewer for the thoughtful comments and have tried to address them accordingly. 

Title, Abstract and Background 

REVIEWER’S COMMENT:

The title, abstract and introduction are adequate. The manuscript brings relevant information about the literature and clearly supports its justification and aims. 

RESPONSE: 

Thank you. 

Methods 

REVIEWER’S COMMENT:

The methods were also well written. I only have a few comments and suggestions for improving the manuscript. 

RESPONSE: 

Thank you. 

REVIEWER’S COMMENT:

Page 7. Line 105. “(…) am attracted"; "7) I'm not attracted to any gender", and; "9) Refuse to answer”.

Comment: The answer options goes from 7 to 9. Is there an option 8 or it was it just a typing error? 

RESPONSE: 

There was no typing error. The investigators designated option 9 on many questions to indicate refusal to answer to facilitate the data entry process. The authors have added the following remarks to the METHODS section: 

" The investigators did not include option 8 on the questionnaire. The investigators designated the value of 9 on many questions to indicate refusal to answer to facilitate the data entry process."

REVIEWER’S COMMENT:

Instrumentation: Measurement of Experience of Violence:

Page 8. Lines 144-146. “We excluded students who had never been in a relationship and students who had never had sex from analyses on intimate partner violence and sexual violence, respectively.”

Suggestion: Excluding participants who reported that never had sex can be misleading. Those participants who answered that they never had sex do not necessarily have not suffered sexual violence, since they may have the understanding that an experience of sexual intercourse is always with consent (different from what happens in rape). So, if it is not feasible to change in the analyses, it would be important to insert this information in the discussion section. 

RESPONSE: 

We thank the reviewer for the comment. We believe that some clarifications may be needed to answer this question, and we have revised our manuscript accordingly. 

Firstly, we did not classify those who never had a romantic partner or never had sex from our analyses. However, we did assume that these individuals did not experience intimate partner violence or sexual violence within the past 12 months prior to the survey. 

The question and answer choices on intimate partner violence was as follow: 

"8. In the past 12 months, has your romantic partner ("แฟน") ever intentionally hit or physically assaulted you?

0) Never

1) Yes

2) Never had a romantic partner

"

And the question and answer choices on sexual assault was as follow: 

"9. In the past 12 months, have you been forced to have sex against your will?

0) Never been forced

1) Yes

2) Never had sex"

Based on this context and the use in the Thai language, and to ease the comparison with the findings of a previous study in Wichaidit et al. (2021), the authors prefer to keep the existing analysis protocol. 

In that regard, the authors have included the following remarks in the METHODS section under measurement of experience of violence: 

"Investigators measured the experience of intimate partner violence with the following question and answer choices: "In the past 12 months, has your romantic partner ("แฟน") ever intentionally hit or physically assaulted you? �0) Never; �1) Yes; �2) Never had a romantic partner". Investigators measured the experience of sexual violence with the following question and answer choices: "In the past 12 months, have you been forced to have sex against your will? �0) Never been forced; �1) Yes; �2) Never had sex". For these questions, we considered students who answered “Never had a romantic partner” and “Never had sex” as those who had never experienced intimate partner violence and sexual violence, respectively. For past-year experience of violence (any type), we treated non-responses regarding each type of violence as missing values. Thus, we only included participants who answered all four questions regarding past-year experience of violence in our analyses." 

The authors also have included the following remarks in the DISCUSSION section: 

" One potentially problematic issue with our study findings was the classification of participants who answered that they never had a romantic partner or never had sex as those who never experienced intimate partner violence or sexual violence, respectively. This classification could be misleading. The definitions of having a romantic partner or having had sex were not included as part of the question, which could have introduced misclassification error in the responses, leading to potential information bias. The questions in our study and the analyses methods eased comparison of the findings with a previous study[16], but should not preclude improvement in future studies." 

Results 

REVIEWER’S COMMENT:

Table 1. I suggest inserting sociodemographic characteristics of the sample in table 1, including information about mean age of the participants in each school year, race and family income. 

RESPONSE: 

The authors thank the reviewer for the comment. However, these characteristics have been included in Table 2a and Table 2b. If possible, the authors wish to keep the characteristics in these tables to avoid redundancy. 

REVIEWER’S COMMENT:

Tables 2a and 2b. The row frequencies of the tables (2a and 2b) do not add up to 100%. Are the weighted percentages the responsible for that? If not, the authors should revise the tables. 

RESPONSE: 

Tables 2a and 2b are part of the same table, split into 2 tables to enable the table to fit into the manuscript’s margins. As such, the totals of the row percent must include those in both tables. The authors have checked and randomly selected row percent totals added to 100 percent. 

No further changes made at this time. 

REVIEWER’S COMMENT:

Page 20. Lines 209-210. “With regards to mental health outcomes and health behaviors, prevalence of depression at time of study and within past year was higher among bisexual, transgender, asexual, and otherwise (…)”.

Comment: This result should be presented with caution. In fact, the study did not assess the presence of depression. The purpose of the PHQ-2 is to screen for depression in a “first-step” approach. Then, the literature suggests that

participants who screen positive (in PHQ-2) should be further evaluated with the PHQ-9 instrument to determine whether they meet criteria for a depressive disorder. So, the PHQ-2 is an instrument that could assess the presence of

symptoms of depression. This information about the difference between presence of disorder and presence of symptoms is important and should be revised in all manuscript.

Suggestion: As the authors compared the continuous data, I suggest to insert information about the means and standard

deviations of the groups in each comparison. 

RESPONSE: 

The authors thank the reviewer for the comment and agree with the point of caution. The authors wish to include the categorization of depressive experience at the time of study as they appeared with the same cut-off point in order to allow for comparison with other studies. However, the authors also agree that continuous data should be present in their original form, and thus included weight means and standard errors of the PHQ-2 score by groups accordingly in Tables 3a and 3b. 

Furthermore, considering that the PHQ-2 measured symptoms of depression, the authors have changed the term to "depressive symptoms" throughout the manuscript.

REVIEWER’S COMMENT:

Table 4b. Part 2. Data on the asexual AFAB group should be reviewed. In table 4, the prevalence for asexual AFAB group is 0% and in the previous paragraph the authors say that "respondents who identified as asexual reported the lowest prevalence of all types of violence”. 

RESPONSE: 

The authors thank the reviewer for the comments. After checking the original data set, the authors wish to confirm that none of the 50 students who were categorized as asexual AFAB by the authors reported experiencing any type of violence within the past year. The authors wish to keep the findings as they appear. 

Discussion

REVIEWER’S COMMENT:

The discussion needs to be improved. The results presented have the potential to be discussed with the current literature on the subject, but the authors need to include more references and discuss with greater depth the studies presented. 

RESPONSE: 

We have tried to revise the DISCUSSION section accordingly.

REVIEWER’S COMMENT:

Limitations are well described, but some appear mixed up throughout the discussion. I suggest that they should be entered under the “Strengths and Limitations” section. The same situation occurs with regard to suggestions that authors make about future studies, these information appear several times throughout the discussion. I suggest to add a paragraph with these suggestions in the limitation section or in the conclusion section. 

RESPONSE: 

We thank the reviewer for the comments and the encouraging remarks. 

We have modified the “Strengths and Limitations” section and added a “Suggestions for Future Studies” sections accordingly. 

REVIEWER’S COMMENT:

The information about the possibility of recall bias when referring to the prevalence of depressive experiences in the last 12 months, could be added as a limitation in the "Strengths and Limitations” section. 

RESPONSE: 

We have added the following remarks to the “Strengths and Limitations” section. However, considering that recall bias is more common in case-control studies, whereas the study data came from a survey (cross-sectional study), the authors decided to refer to the phenomenon as "outcome misclassification" of study participants instead.

REVIEWER’S COMMENT:

Page 30. Lines 254-255. "Such prevalence is higher than a previously-estimated 8 percent prevalence [13], and is higher than the global average even when those who provided incomplete information were taken into account[14]

Suggestion 1: Please specify the global average of the cited study. Also, if possible, present studies from different countries.

Suggestion 2: It may be important to bring more information on the mental health of young people who identify as LGBTQIA+, for example, citing Meyer's minority stress theory. I also suggest discussing other studies on mental health, substance abuse, sexual behavior and exposure to violence. 

RESPONSE: 

Changes made as per Suggestion 1 to the sentence as follows: 

“Such prevalence is higher than a previously-estimated 8 percent prevalence [13], and is higher than the global average of 11 percent even when those who provided incomplete information were taken into account[14].”

Remarks added to the DISCUSSION section as per Suggestion 2: 

“Meyer's Minority Stress model offers a theoretical framework that stigma, prejudice, and discrimination faced by LGBTQA+ people "create a hostile and stressful social environment that causes mental health problems"[30], albeit the model may need to be further modified[31]. A survey in Australia among young people attracted to those of the same sex showed that internalized homophobia, perceived stigma, and experiences of homophobic physical abuse were associated with suicidal thoughts[32]. It is possible that adolescent LGBTQA+ individuals in our study were also subject to similar stressors, such as sexual/gender stigma [29] and internalized discrimination[33], which then influenced their suicidality.”

REVIEWER’S COMMENT:

Page 32. Lines 301-303. "The higher prevalence of drinking in certain groups could be a reflection of unhealthy coping mechanisms[26], or a reflection of broader societal trends [27]."

Suggestion: Please, insert the mentioned societal trends. 

RESPONSE: 

Added remark “…in which alcohol consumption is declining among adolescents and youths”

REVIEWER’S COMMENT:

Page 32. Lines 303-306. "Such disparities were also observed for lifetime history of illicit drug use, although homosexual male participants seemed to have notably higher prevalence in nearly all types of substances compared to other groups, which differed from a similar study elsewhere [28]”.

Suggestion: Please, give more information such as location and data about the cited study.

RESPONSE: 

The following remark has been added as per the comment: 

“This differed from the findings of a national survey on drug use in the United States, which found that bisexual women and bisexual men (in addition to gay men) had 2-3 times higher prevalence of substance use compared to heterosexual adults[28].”

################################################

Reviewer #2 

REVIEWER’S COMMENT:

I was enthusiastic about reading this paper on sexual attraction and gender identity-based differences in health among adolescents in Thailand. However, I think the paper can be approved on. 

RESPONSE: 

The authors thank the reviewer for the comments and have tried to address them accordingly. 

REVIEWER’S COMMENT:

There is not a lot of rationale as to why to study LGBTQ+ adolescents. I am also not convinced by the operationalization of the LGBTQ+ groups. Because of this, I am not sure how valuable these findings really are or how they should be interpreted. They authors should come up with a strong rationale for this operationalization or change it. See below my comments. 

RESPONSE: 

The authors thank the reviewer for the helpful comments. With regard to the rationale for the study, we hope that study findings will contribute empirical evidence on health disparities between LGBTQ+ groups vs. non-LGBTQ+ individuals. Such findings can help identify groups that are particularly vulnerable and inform resource allocation accordingly. If disparities occurred in a pattern, the findings can generate hypotheses for further studies. 

We have tried to make changes to the manuscript accordingly. 

Abstract 

REVIEWER’S COMMENT:

1. The term “cisgender-heteronormative persons” reads awkward to me. I believe it is more common to use a term like “heterosexual or cisgender people”. I would suggest to change this throughout. 

RESPONSE: 

We thank the reviewer for the comment. The authors deem the term “heterosexual” to refer to sexuality, whereas “cisgender” to refer to gender identity being the same as gender assigned at birth. However, the authors also recognize that the term “cisgender-heteronormative” implies an extent of conformity to existing social norms, which were not measured. Therefore, the authors decide to replace the term “cisgender-heteronormative” with “cisgender-heterosexual”, but keep the latter throughout the manuscript. 

REVIEWER’S COMMENT:

2. It is not clear from the background section that the study focuses on adolescents. Some rationale is needed why studying sexual and gender identity-based health disparities specifically among adolescents is needed. 

RESPONSE: 

We have included the following remarks in the INTRODUCTION section:

“Adolescence is the period in which people begin to develop a full understanding of their own gender identity and sexual orientation, but is also the age group where behavioral health issues impose a heavier burden relative to other health problems.” 

REVIEWER’S COMMENT:

3. In the methods or results section, it would be helpful to also report the mean age and standard deviation of the sample. 

RESPONSE: 

We thank the reviewer for the comment. As age is correlated with year of study in secondary education, we would prefer to report the year of attendance and school system rather than the age at the time of survey. In that regard, we have added the following remarks to the Results section of the Abstract: 

" Participants who identified as LGBTQA+ were more likely to be in older year levels "

REVIEWER’S COMMENT:

4. Violence is mentioned throughout the abstract, but I am not sure what is meant by this. Is this physical violence, verbal violence? Is it akin to discrimination? RESPONSE: 

The term “experiences of violence” refers to the self-reported experiences of violence within 12 months prior to the survey, including physical or verbal violence victimization (with involvement of a weapon), altercation with others (with injuries requiring medical treatment), intimate partner violence, and sexual violence

Changes made throughout the manuscript

REVIEWER’S COMMENT:

5. In the first sentence of the “Results” section of the abstract, the number is 23,659 is mentioned twice, which is unnecessary 

RESPONSE: 

The second mentioning of the number has been removed. 

REVIEWER’S COMMENT:

6. It is first stated that 23,659 participants returned complete and valid questionnaires. Then it stated that “10 percent provided incomplete information with regard to gender identity and sexual orientation”. I do not understand this. 

RESPONSE: 

Among the 23,659 participants who answered at least 70 percent of the questions without skip pattern, approximately 9.6% did not answer all 3 questions required to identify a student as LGBTQA+ or non-LGBTQA+. 

In order to avoid confusion, we have decided to remove the remark from the Results section of the Abstract

Introduction 

REVIEWER’S COMMENT:

7. I am not sure whether I agree with the definition following definition: “the term "transgender" refers to the state in which an individual's gender identity is different from their sex assigned at birth.” Someone who identifies as nonbinary has a different gender identity then the assigned sex at birth, but is not considered transgender. Also, the word “state” reads awkward here. I would suggest to search for references in which more correct definitions are used. The same holds for the definition of sexual orientation, which also seems incomplete to me. 

RESPONSE: 

According to the Human Rights Campaign Foundation in collaboration with the American College of Osteopathic Pediatricians and the American Academy of Pediatrics, the term "transgender" is an umbrella term that refer to someone whose gender identity does not match their sex assigned at birth, and can be used to refer to "transgender girls, transgender boys, and non-binary people" [1]. The authors of this manuscript chose to use the term "state" in order to avoid using the term "transgender" as a noun, but rather as an adjective, in order to comply to the preference of stakeholders in LGBTQA+ issues, such as the Gay and Lesbian Alliance Against Defamation (GLAAD) [2]. However, to ease the communication process, the authors have now opted to use the definitions in quotations. 

INTRODUCTION section extensively revised. 

References: 

1. Human Rights Campaign Foundation, American College of Osteopathic Pediatricians, American Academy of Pediatrics. Supporting & Caring for Transgender Children [Internet]. 2016 [cited 2023 Apr 25]. Available from: https://assets2.hrc.org/files/documents/SupportingCaringforTransChildren.pdf

2. Gay and Lesbian Alliance Against Defamation (GLAAD). GLAAD Media Reference Guide 11th Edition [Internet]. Gay and Lesbian Alliance Against Defamation (GLAAD). 2022 [cited 2023 Apr 25]. Available from: https://www.glaad.org/reference/transgender

REVIEWER’S COMMENT:

8. I do not understand the sentence “(LGBTQA+) refers to any person who does not identify as cisgender-heteronormative, regardless of their gender identity, sexual orientation, or asexuality”. If you do not identify as “cisgender-heteronomrative” than you will probably identify as LGBTQ+, so I do not understand the “regardless of their gender identity, sexual orientation, or asexuality” part of this definition. Also, when someone does identify as cisgender-heteronormative, but is somewhat attracted to people of the same gender, how would they then be categorized? The authors should put more effort in providing a clear description of what they mean with terms they are using. Right now, it seems there is only a surface level understanding of sexual identity, behavior, attraction and gender identity. These concepts overlap, but are not the same and while reading this, I do not get a sense that authors understand these differences. Asexuality is further often considered as a sexual orientation, singling is out here seems incorrect. 

RESPONSE: 

The remark ", regardless of their gender identity, sexual orientation, or asexuality" was meant to convey the idea that if one does not identify as cisgender-heteronormative, then one will probably identify as LGBTQA+. However, the wording appeared to have created confusion. The remark has been removed from the INTRODUCTION section. The phrase "regardless of..." has also been removed from Table 1 in order to avoid similar confusions.

With regard to the categorization, if the study participant reported being attracted to more than one gender in question H1b, the participant would be classified as being "bisexual/polysexual". The question H1b was worded "To which gender are you attracted? (multiple answers allowed)", and thus did not capture the intensity of the attraction itself.

The authors have attempted to provide a clear description of the categorization in Table 1 as well as the annotations in the R codes in the supplementary materials. The authors acknowledge that the categorization in Table 1 was a very crude and superficial way to describe people. Unfortunately, without resorting to such methods, it would be very difficult to estimate the proportion of Thai adolescents who identified as non-cisgender-heteronormative and to describe the existing health disparities. 

The authors wish to hereby thank the reviewer for the thoughtful comments and will attempt to revise the INTRODUCTION section where possible. The authors also hope to receive further comments from the reviewer in the next iteration of the manuscript, if at all possible. 

REVIEWER’S COMMENT:

9. What is meant with “social and human rights issues”? Similarly, what do you mean with violence and abuse? Further, more detail is needed when discussing findings of other (empirical?) studies.

RESPONSE: 

We have revised the remarks to " LGBTQA+ people tend to have poorer behavioral health[8,9] and face greater level of physical and sexual violence than cisgender-heterosexual people[10,11].".

REVIEWER’S COMMENT:

10. On line 64, your mention health disparities. Compared to whom have research identified these disparities? 

RESPONSE: 

We have revised the remarks to " LGBTQA+ people tend to have poorer behavioral health[8,9] and face greater level of physical and sexual violence than cisgender-heterosexual people[10,11].".

REVIEWER’S COMMENT:

11. I think a little more attention can be paid to that most research on LGBT+ youth is not conducted in Thailand. I am also not sure whether the few studies conducted in Thailand focus on adolescents. It should also be made clearer why it is crucial to study these disparities during adolescence. What precisely is the research gap? 

RESPONSE: 

INTRODUCTION section revised accordingly. 

REVIEWER’S COMMENT:

12. Last, I am also not sure about the whole structure of the introduction. Only form the third paragraph onward it becomes clear what the goal of this study is. You should try to revise the structure of this section to make it clear from the start what you are studying, what the research question is, and who the population is. 

RESPONSE: 

We have attempted to revise the structure and content of the INTRODUCTION section to allow for greater clarity and better flow of ideas. 

Materials and Methods 

REVIEWER’S COMMENT:

13. When discussing the sample, make sure to mention the typical ages of the participants. I am not sure how old students in year 7, 9 and 11 are. I am also wondering why 8 and 10 were not included in the study. 

RESPONSE: 

We chose students in years 7, 9 and 11 in order to cover a wide an age range as possible. Excluding years 8 and 10 allows us to avoid potential overlaps in age between adjacent years of study. 

We have included the following remarks in the METHODS section: 

" Thailand follows the 6-3-3 education system and all students begin Year 1 (Prathom 1) during the year that they are to reach 7 years of age[17]. The system requires students to repeat a grade level only in the most extreme circumstances[18]. Thus, the majority of participants in Year 7 were 12-13 years of age, participants in Year 8 were 14-15 years of age, and participants in Year 11 were 16-17 years of age. We included all students present in the sampled classroom on the day of data collection. We excluded students in Year 8, 10, and 12 in order for our study data to cover as broad an age range as possible given the existing statistical power."

REVIEWER’S COMMENT:

14. I am also wondering how the survey was administered (paper pencil during a class?), if participation was voluntary, and whether consent was obtained? RESPONSE: 

The authors have included additional details in the Procedure sub-section of MATERIALS AND METHODS. 

REVIEWER’S COMMENT:

15. The sentence “We used the responses of three questions to define the genders in this study” is awkward. More correct would be to discuss that three questions were used to assess gender identity. It is also incorrect that questions on sexual attraction are used to inform gender identity. Again, a better understanding is needed of how gender identity and sexual orientation are distinct. 

RESPONSE: 

We apologize for our mistake. We meant to convey the notion that "We used two questions to assess gender identity and one question to assess sexual orientation." Additional corrections made to the Instrumentation: Measurement of Gender and Sexual Orientation sub-section of MATERIALS AND METHODS.

REVIEWER’S COMMENT:

16. It is irrelevant to mention from which sections of the questionnaire certain question came from. 

RESPONSE: 

Noted. Changes made. 

REVIEWER’S COMMENT:

17. When discussing questions, it would be helpful if a translated question is presented, next to the already present answer options. 

RESPONSE: 

We have included a full unofficial translation of the study questionnaire as a supplementary material upon the submission of this revised version of the manuscript. 

REVIEWER’S COMMENT:

18. I am also not sure about the sentence “We used the responses of three questions to define the genders in this study.” You are also mentioning sexual attraction questions here, which is different from gender identity. This sentence does not reflect this. In general, I am not sure why gender identity and sexual attraction were combined this way. What is the relevance/rationale behind this? What do we gain from using this operationalization? 

RESPONSE: 

We have revised the Instrumentation: Measurement of Gender and Sexual Orientation sub-section of MATERIALS AND METHODS in an attempt to clarify this matter. Changes made. 

REVIEWER’S COMMENT:

19. Sexual attraction is measured in this study. However, participants are described as being heterosexual and gay. But these identities were not measured. I would be more correct to refer to your participants not by using identity labels, but by their attractions (e.g., other gender attracted, same gender attracted). RESPONSE: 

We thank the reviewer for the comment and we have noted the reviewer’s concern. We have revised Table 1 accordingly. 

After internal deliberations, we decided to change the Group description for cisgender homosexual boys to: 

“3) Cisgender boys, attracted only to cisgender boys (“Cisgender Homosexual Boy” or “Gay”) (n=496 respondents)”

We have also decided to change the Group description for cisgender homosexual girls to: 

“4) Cisgender girls, attracted only to cisgender girls (“Cisgender Homosexual Girl” or “Lesbian”) (n=619 respondents)”

We also noted the subjectivity of the terms in the table footer as follows: 

“The labels in the quotation marks in the Group column are based only on the subjective interpretation of the authors”

REVIEWER’S COMMENT:

20. I also have difficulty understanding why some someone who is “Cisgender Homosexual (“Gay”) Boys” is someone who is only attracted to cis-gender boys. If someone is attracted to cisgender boys and transgender boys, they can be considered gay, because they are attracted to men. Why did the authors make this distinction? 

RESPONSE: 

We thank the reviewer for the thoughtful comment. In this study, we considered all cisgender persons who reported being attracted to more than one genders as being polysexual, and those who did not fit the definition of being in groups 1 thru 6 in Table 1 (“cisgender heterosexual boys”, “cisgender heterosexual girls”, “cisgender homosexual boys”, “cisgender homosexual girls”, “bisexual/polysexual boys”, “bisexual polysexual girls”) but do not identify as transgender or asexual to be “queer and questioning”. 

In that regard, a cisgender male participant who reported being attracted to cisgender boys and transgender boys will be considered by our classification system as being “bisexual/polysexual boys” simply because they reported being attracted to more than one identity. 

We acknowledge the narrow nature of these definitions, but we deemed this to be the only way to start describing the diversity of Thai youths with regard to both gender identity and sexuality. We remain open to additional revisions and hereby welcome the reviewer’s further suggestions. 

REVIEWER’S COMMENT:

21. Why are depressive experience and suicidality considered as behavioral health? 

RESPONSE: 

According to the American Medical Association, behavioral health refers to the "prevention, diagnosis and treatment" of "mental health and substance use disorders, life stressors and crises, and stress-related physical symptoms" (https://www.ama-assn.org/delivering-care/public-health/what-behavioral-health). We deemed depressive experience to be a symptom of mental health disorder, and suicidality to be a life crisis. Thus, we considered those two outcome domains to be parts of behavioral health.

REVIEWER’S COMMENT:

22. For depressive experience, can you provide a sample question and the answer categories and how one score was obtained? 

RESPONSE: 

We thank the reviewer for the comment. We have included the translated version of parts of the study questionnaire in the Supplementary Material section. We have decided to refrain from adding details to this segment in order to avoid redundancy. 

REVIEWER’S COMMENT:

23. For sexual activity, you mention that “questions were largely similar to the previous round of study”. I do not know what this refers to, as I am not familiar with the “previous round”. Is this information really needed? It is also mentioned here that sexual orientation was measured. Why was this not used to measure sexual orientation, but was sexual attraction used instead? 

RESPONSE: 

We thank the reviewer for the comment. We do not think that such information is needed, and the remark has been removed in order to avoid confusion. 

We also have made additional revisions in an effort to improve clarity. 

REVIEWER’S COMMENT:

24. For drinking, tobacco, and drug use, please provide sample questions and answer options and describe more precisely how measures were constructed. RESPONSE: 

We thank the reviewer for the comment. We have included the translated version of parts of the study questionnaire in the Supplementary Material section. We have decided to refrain from adding details to this segment in order to avoid redundancy. 

REVIEWER’S COMMENT:

25. For experiences with violence, what were the answer options? 

RESPONSE: 

We thank the reviewer for the comment. We have included the translated version of parts of the study questionnaire in the Supplementary Material section. We have decided to refrain from adding details to this segment in order to avoid redundancy. 

REVIEWER’S COMMENT:

26. The “Procedure” section should be moved up to the beginning of the Materials and Methods section.

RESPONSE: 

Change made as per the reviewer’s suggestion

REVIEWER’S COMMENT:

27. In the data analysis section, weekly allowance is mentioned. This wat not mentioned in the measurement section, I am not sure what this is referring to. RESPONSE: 

Changes made as follows: 

“We compared continuous data (i.e., mean amount of weekly allowance reported by participants in each group in Table 2) using one-way ANOVA.”

REVIEWER’S COMMENT:

28. Did the authors also look into missing data mechanisms. Was the data, for instance, missing at random?

RESPONSE: 

We thank the reviewer for the suggestion. We have compared the characteristics of respondents who reported complete vs. incomplete information on gender and sexuality as Supplementary Table 1 in the supplementary material section and revised the content of the METHODS and RESULTS section accordingly. 

REVIEWER’S COMMENT:

29. The “Ethical Considerations” section should be moved up to the beginning of the Materials and Methods section. 

RESPONSE: 

Change made as per the reviewer’s suggestion

REVIEWER’S COMMENT:

30. In general, why are only bivariate associations considered? Why are no control variables introduced to the models? This would make a more convincing paper.

RESPONSE: 

We thank the reviewer for the comment. We did not perform multivariate analyses in this manuscript simply because we were concerned about the use of manuscript space given the relatively large number of comparison groups. 

I would like to hereby mention a similar study published in PLOS One using data from a previous round of the Survey:

Wichaidit W, Assanangkornchai S, Chongsuvivatwong V. Disparities in behavioral health and experience of violence between cisgender and transgender Thai adolescents. PLoS One. 2021;16(5):e0252520. Published 2021 May 28. doi:10.1371/journal.pone.0252520 

In that previous study, there were 6 comparison groups, and the authors compared the outcomes between each non-cisgender group against cisgender boys and cisgender girls separately. Thus, for each comparison group, there were two adjusted odds ratios calculated (one against cisgender male participants, and the other against cisgender female participants). 

With the existing bivariate format, each of our comparison tables are already split into two parts across multiple pages in order to accommodate to the breadth of the content with 12 comparison groups. To follow a similar strategy to the previous study in our analyses (with “cisgender heterosexual boys” and “cisgender heterosexual girls” as the reference groups), we would need 20 adjusted ORs (thus 22 additional columns) to be calculated separately for each outcome, which would stretch the breadth of the table even further. Thus, we would prefer to keep the existing bivariate format. 

Results 

REVIEWER’S COMMENT:

31. Again, not sure what you mean by complete questionaries when also incomplete questions are mentioned.

RESPONSE: 

The exclusion was with regard to the exclusion of participants who were deemed to have submitted incomplete questionnaires and excluded from the analyses, as mentioned in the METHODS section as follows: 

“Participants who answered less than 70 percent of the questions in which skip patterns did not apply were considered to have submitted incomplete responses and were excluded from the analyses.” 

However, in order to avoid confusion, we have revised the opening sentence of the RESULTS section as follows: 

“There were 24,143 participants who placed their questionnaires in the envelope, among whom 23,659 (98.0%) were deemed to have filled the questionnaires adequately and were included in our analyses.”

REVIEWER’S COMMENT:

32. On page 13, control variables are mentioned that were not introduced in the matarials and method section. 

RESPONSE: 

We are not sure of the segment to which the reviewer refers. We apologize for not being able to address this comment. In that regard, none of our analyses had control variables.

REVIEWER’S COMMENT:

33. It is mentioned that “Generally, students at government schools and those in Matthayom 3 and Matthayom 5 more commonly identified as LGBTQA+ than students at private schools and students in Matthayom 1 or vocational schools.” Why is this the case? 

RESPONSE: 

Remarks regarding these differences added to the DISCUSSION section at the end of the second paragraph. 

REVIEWER’S COMMENT:

34. I have a lot of difficulty really understanding the results. This is mainly because the “gender identity groups” are not in line with previous research and therefore hard to interpret what these differences mean. I strongly advise to change this. 

RESPONSE: 

We thank the reviewer for the comment. We have tried to revise the manuscript accordingly. 

REVIEWER’S COMMENT:

35. You should mention in the tables what the referent category is. It is hard to read the tables now.

RESPONSE: 

We thank the reviewer for the comment. To clarify, our analyses did not include logistic regression or variations thereof. The p-values in Tables 2 thru 4 were mainly from Chi-square test of association. Thus, there were no referent categories. 

In that regard, to avoid the manuscript from becoming misleading, we have revised the title of our paper accordingly.

Discussion 

REVIEWER’S COMMENT:

36. I do not think that a sentence like “We found that LGBTQA+ youths overall had higher prevalence of depression, suicidality, lifetime sexual activity, alcohol and tobacco use, lifetime history of illicit drug use, and past-year exposure to violence” should be included, as you mention directly afterwards that there is heterogeneity. By stating this, the heterogeneity is not paid attention to. In general, more attention should be given to these findings and what the implications are.

RESPONSE: 

We thank the reviewer for the comments. Revisions made. 

REVIEWER’S COMMENT:

37. When discussing the high numbers of LGBTQ participants, I think that you should also refer to other studies that found high prevalence among Gen Z compared to previous generations.

RESPONSE: 

Remarks added with citation to the DISCUSSION section. 

REVIEWER’S COMMENT:

38. In general, more effort should be put into referring to previous studies in understanding the current results. For instance, it is mentioned that “Teenagers who initially identify as transgender may, later in their adolescence, identified as gender diverse and decided not to undergo the transitioning process”. Does this happen often? Is there any information on this?

RESPONSE: 

The quoted remark has been removed from the DISCUSSION section. 

REVIEWER’S COMMENT:

39. When making recommendations on gender identity questions it is recommended to use more inclusive terms. Is this in general a recommendation, or specifically for the Thai context? 

RESPONSE: 

Our recommendations were specifically in the Thai context. We have included the remark “specifically in the Thai context” in the "Suggestions for Future Studies" sub-section accordingly.

#######################################################

Reviewer #3 

REVIEWER’S COMMENT:

This study is based on the 5th National School Survey on Alcohol Consumption, Substance Use, and Other Health-Risk Behaviors. This study sought to examine how behavioral health outcomes and exposure to violence differed between cisgender-heterosexual youths and LGBTQA+ youths. 

RESPONSE: 

(No response)

Major comments 

REVIEWER’S COMMENT:

Q1. Suggested authors extended the concept of 1)disparities in Behavioral Health and 2) experience of Violence between CisgenderHeterosexual comprises Lesbian, Gay, Bisexual, Transgender, Queer and Questioning, and Asexual (LGBTQA+), and 3) the reason choose Thai Adolescents 

RESPONSE: 

We have revised the INTRODUCTION section and tried to address these issues accordingly. 

REVIEWER’S COMMENT:

Q2. Suggested the wording in the abstract is within 250 words. 

RESPONSE: 

We have checked the journal’s requirement and it seems that there is no word limit. However, we have revised the abstract to reduce the length and make the content more concise. 

REVIEWER’S COMMENT:

Q3. The introduction has some literature review content, so a section for literature review should be created. 

RESPONSE: 

We thank the reviewer for the comment. We have revised the INTRODUCTION section and has included more literatures, which we hope to function in a similar manner to a literature review section. 

REVIEWER’S COMMENT:

Q4. Suggested to rewrite 82-147, the instrumentation section needs to be shorter and specifically highlight how the instrument is a good fit with this study. RESPONSE: 

We have revised the Instrumentation sections accordingly. We have also provided a partial translation of the questionnaire in the supplementary material section. Unfortunately, in order to respond to comments from other reviewers, we could not shorten the section. 

REVIEWER’S COMMENT:

Q5. Suggest adding the statistically analyze section. 

RESPONSE: 

The "Statistical Analysis" sub-section has been added to the METHODS section

REVIEWER’S COMMENT:

Q6. In line 195, a) even the authors draw the results Percent ± SE*; however, what are the results under meanings behind? b) what is the comparison of the results from all groups? 

RESPONSE: 

We have included the following remarks at the end of the "Statistical Analysis" sub-section in METHODS: 

" We presented prevalence data as weighted percentage ± standard error (SE), the latter of which can be regarded as a margin of potential sampling error."

REVIEWER’S COMMENT:

Q7. Suggested clear Inclusion and exclusion criteria with the paragraph. 

RESPONSE: 

We have added information regarding inclusion and exclusion of study participants as the last two sentences of the "Study Design and Participants" sub-section in METHODS.

REVIEWER’S COMMENT:

Q8. What are the new insights from this paper, and how would the author suggest adding a section on future implications and limitations? 

RESPONSE: 

We have added a "Suggestions for Future Studies" sub-section at the end of DISCUSSION accordingly.

REVIEWER’S COMMENT:

Q9. When the results are presented in the result section, how do you consider the results significant to the (LGBTQA+)? 

RESPONSE: 

We have checked our narration of the study findings in RESULTS to make sure that we focused on disparities between comparison groups accordingly. 

REVIEWER’S COMMENT:

Q10. Please provide background information concerning LGBTQA+ differences in behavioral health and experiences of violence to support teens and compare them with adults, even older adults. 

RESPONSE: 

We have extensively revised the INTRODUCTION section to provide a background regarding the LGBTQA+ identities, and also cited the following reference which investigated similar issues to those in the current manuscript

Wichaidit W, Assanangkornchai S, Chongsuvivatwong V. Disparities in behavioral health and experience of violence between cisgender and transgender Thai adolescents. PLoS One. 2021;16(5):e0252520. Published 2021 May 28. doi:10.1371/journal.pone.0252520 

REVIEWER’S COMMENT:

Q11. Suggested double check and explain the results in table 1 group 9 and 10 0.3%±0.0% and 0.2%±0.0%, the ±0.0% what is the implication? RESPONSE: 

We thank the reviewer for the comments. To clarify, there were very few participants (relative to the entire group) who identified as asexual (approximately 0.2% to 0.3%). With these low prevalences, the standard errors (as margin of errors in the approximation) were less than 0.1%, and thus were rounded as "0.0%" when the decimals were rounded to one digit.

REVIEWER’S COMMENT:

Q12. Family house/flat (n=18,461), p-value 0.115, what is the meaning of p-value when it refers to the mental health and violence experience of the teenage LGBTQA+ group? 

RESPONSE: 

We thank the reviewer for the comment. The p-value simply denote that there were no statistically significant differences with regard to living situation between comparison groups in our study.

REVIEWER’S COMMENT:

Q13. The revised manuscript suggested citing the relevant reference in the following papers.

doi: 10.3389/fpsyg.2021.677734; doi:15579883221120985.; doi: 10.3389/fpsyg.2022.726343; doi: 10.3389/fpsyg.2021.704995; doi: 10.3389/fpsyg.2021.692343

The suggested author submits the manuscript to the editing service to ensure the manuscript meets the requirement of language quality. 

RESPONSE: 

We have included the following research and review papers into the INTRODUCTION section: doi:15579883221120985; 

doi: 10.3389/fpsyg.2022.726343

---

## [Decision Letter · Decision Letter 1]

31 May 2023

Behavioral Health and Experience of Violence among Cisgender Heterosexual and Lesbian, Gay, Bisexual, Transgender, Queer and Questioning, and Asexual (LGBTQA+) Adolescents in Thailand

PONE-D-22-31754R1

Dear Dr. Wichaidit,

We’re pleased to inform you that your manuscript has been judged scientifically suitable for publication and will be formally accepted for publication once it meets all outstanding technical requirements.

Kind regards,

Marianna Mazza

Academic Editor

PLOS ONE

Additional Editor Comments (optional):

Reviewers' comments:

Reviewer's Responses to Questions

**Comments to the Author**

1. If the authors have adequately addressed your comments raised in a previous round of review and you feel that this manuscript is now acceptable for publication, you may indicate that here to bypass the “Comments to the Author” section, enter your conflict of interest statement in the “Confidential to Editor” section, and submit your "Accept" recommendation.

Reviewer #1: All comments have been addressed

Reviewer #3: All comments have been addressed

2. Is the manuscript technically sound, and do the data support the conclusions?

Reviewer #1: Yes

Reviewer #3: Yes

3. Has the statistical analysis been performed appropriately and rigorously? 

Reviewer #1: Yes

Reviewer #3: Yes

4. Have the authors made all data underlying the findings in their manuscript fully available?

Reviewer #1: Yes

Reviewer #3: Yes

5. Is the manuscript presented in an intelligible fashion and written in standard English?

Reviewer #1: Yes

Reviewer #3: Yes

6. Review Comments to the Author

Reviewer #1: The manuscript brings relevant information about the literature and clearly supports its justification and aims. Also, the study is relevant for publication and highlights important information about the health-disease process of LGBTQA youth and also their health disparities. The paper approaches a relevant topic, the authors use appropriate weighting procedures to overcome the complex design and presented data from a large national school survey. After the revision, the manuscript appears to be ready for publication. I agreed with all answers and corrections provided by the authors.

Reviewer #3: I appreciate the author's efforts and am eagerly anticipating the publication of this paper. I have no additional follow-up to provide for this manuscript.

7. PLOS authors have the option to publish the peer review history of their article (what does this mean?). If published, this will include your full peer review and any attached files.

Reviewer #1: No

Reviewer #3: **Yes: **Alex Siu Wing Chan

---

## [Editor Report · Acceptance letter]

5 Jun 2023

PONE-D-22-31754R1 

Behavioral Health and Experience of Violence among Cisgender Heterosexual and Lesbian, Gay, Bisexual, Transgender, Queer and Questioning, and Asexual (LGBTQA+) Adolescents in Thailand 

Dear Dr. Wichaidit:

I'm pleased to inform you that your manuscript has been deemed suitable for publication in PLOS ONE. Congratulations! Your manuscript is now with our production department. 

Kind regards, 

on behalf of

Dr. Marianna Mazza 

Academic Editor

PLOS ONE